# A non-monotonic code for event probability in the human brain

Cedric Foucault [1,2,5] ✉, Tiffany Bounmy[1,5], Sébastien Demortain[1], Bertrand Thirion [3], Evelyn Eger[1] & Florent Meyniel [1,4] ✉

Assessing probabilities and predicting future events are fundamental for perception and adaptive behavior, yet the neural representations of probability remain elusive. While previous studies have shown that neural activity in several brain regions correlates with probability-related factors such as surprise and uncertainty, similar correlations have not been found for probability. Here, using 7 Tesla functional magnetic resonance imaging, we uncover a representation of the probability of the next event in a sequence within the human dorsolateral prefrontal and intraparietal cortices. Crucially, univariate and multivariate analyses revealed that this representation employs a highly non-monotonic code. Tuning curves for probability exhibit selectivity to various probability ranges, while the code for confidence accompanying these estimates is predominantly monotonic. Given such diversity in tuning curves, future studies should move from assuming monotonic or simple canonical forms of tuning curves to considering richer representations, and clarify why different types of code exist.

Our world is conveniently described in terms of probabilities. These probabilities can be about many different things and concern different aspects of cognition. For example, a probability can be about the veracity of a perception, (e.g., "I'm sure I saw someone I know in the crowd"[1]), a judgment in the context of a social interaction ("My boss is probably lying about my pay rise"[2]), the occurrence of some future event inferred by reasoning or learning ("It is unlikely to rain tomorrow"[3]). These probabilities reflect the uncertainty of our beliefs, due to the ambiguity of our inputs or limitations in our information processing capabilities[4]. Human behavior is adapted to these probabilities, as shown by numerous examples both in laboratory experiments and in real life. To name a few, perception adapts to the probability of occurrence of a given object or feature in the visual, auditory, and somatosensory spaces[5], and choices are guided by the probability of reward[6]. These behavioral results imply that probabilities must be encoded in the brain.

The current literature on the neural code of probability is mostly restricted to reward probabilities[7], the probability of a decision being correct[8], and the related notion of evidence supporting a decision[3]. These probabilities have in common that they have a valence, with higher probabilities (of reward, of being correct) being generally more desirable. In comparison, the neural code for probabilities about affectively neutral events remains largely unknown. Examples are extremely diverse, including the probability of the next word in a sentence, of the temperature increasing tomorrow, of hearing a dog barking at night. Such probabilities are essential to build a rich internal model of our environment. In what follows we refer to the probability of an affectively neural event, which is our focus, simply as "probability".

Several studies have reported that neural activity increases (or decreases) monotonically with quantities that are related to probabilities, but not with probabilities themselves. Examples of related quantities include the coding of prediction error[9], surprise[10], and predictability[11]. Some studies explicitly reported a failure to identify linear correlations between probabilities and neural activity[12,13].

[1]Cognitive Neuroimaging Unit, NeuroSpin (INSERM-CEA), University of Paris-Saclay, Gif-sur-Yvette, France. [2]Sorbonne University, Doctoral College, Paris, France. [3]Inria, CEA, University of Paris-Saclay, Palaiseau, France. [4]GHU Paris, psychiatrie et neurosciences, Hôpital Sainte-Anne, Institut de neuromodulation, Paris, France. [5]These authors contributed equally: Cedric Foucault, Tiffany Bounmy. ✉e-mail: cedric.foucault@gmail.com; florent.meyniel@cea.fr

Here, we tested the possibility that neural activity does not scale with probability levels, but instead that the neural code for probability is non-monotonic. The codes we studied belong to the broader class of codes that are characterized by tuning curves. A tuning curve relates the value of a variable (e.g., the orientation of a grating[14], the number of items in a set[15,16]) to the neural activity that this value elicits on average across trials. A special case of tuning curve is the smooth monotonic code, where neural activity smoothly increases along the encoded quantity. The monotonic code has been extensively tested, in particular in neuroimaging, and it accounts for the encoding of various quantities such as prediction error, surprise, predictability as mentioned above, and other quantities not related to probabilities, e.g., value[17], complexity[18], salience[19]. There are also numerous examples of quantities that are encoded with a non-monotonic code, such as orientation[14], numerosity[16], proportion[20]. For these quantities, tuning curves are highly non-monotonic. They are often bell-shaped, but more complex tuning curves have also been observed[21,22]. In the following, we make a distinction between a smooth monotonic code (possibly linear) and smooth non-monotonic codes.

To adjudicate between monotonic and non-monotonic codes for probability, it is necessary to characterize the brain's tuning curves for probability. We measured neural activity with ultra-high field fMRI (7T) because it provides a high signal-to-noise ratio to capture the neural code at a millimeter-scale noninvasively (see Discussion for implications at the single-neuron level), with a whole-brain coverage (which is very useful given the lack of anatomical priors on the brain regions involved in the representation of probabilities). We characterized tuning curves for probability in each fMRI voxel using a method that combines the generality of functional approximation with basis functions[23] and encoding models[24,25]. This method, which we detail below, makes minimal assumptions about the tuning shape, in contrast to alternative encoding methods (see Discussion[26,27,]). It is also more informative about tuning curves than decoding methods. For example, linear classifiers (e.g., regularized linear regression[28], support vector machine[25]) are often used to decode a quantity from brain signals, but these methods can achieve high performance regardless of whether the underlying tuning curves are monotonic, bell-shaped, or even more complex[29]. In this study, we quantified the form and degree of non-monotonicity of tuning curves at the level of single voxels (following a univariate approach) to adjudicate between monotonic and non-monotonic code.

The monotonic vs. non-monotonic nature of the code also makes specific predictions about the geometry of the activity patterns that code for probability across voxels. To strengthen our conclusions on the code of probability, we also characterized this geometry with a multivariate approach. More precisely, we quantified the similarity of patterns of voxel responses across probabilities, and tested whether it conformed to a monotonic or a non-monotonic code.

In terms of experimental design, we focused on the probability of occurrence of an (affectively) neutral event in a sensory sequence. In making this choice, we do not assume that all types of probability (including, e.g., reward probability) share the same code; in fact, the method we propose here for distinguishing between monotonic and non-monotonic codes can be applied in other contexts to address this question. Our choice is motivated by the idea that studying the neural representation of some quantity requires minimizing potential confounds between this quantity and other constructs and task features[4]. For example, in several previous studies the concept of probabilities has been confounded with evidence for a decision[3], or with the reward that the participant expects to receive[30]. Probabilities are also correlated with some motor or sensorimotor transformations in many tasks[3], which is useful for studying the behavioral relevance of probabilities, but can be a nuisance when studying their code. To alleviate the problem of confounding factors and variables, we used a probability learning paradigm in which participants estimated the (changing) generative probability of occurrence of affectively neutral items presented sequentially. This paradigm efficiently samples a large number of probability estimates, as the latent probability is learned sequentially and is changing. To limit confounds related to motor actions and decision making, this estimation was covert on most trials (on which the analysis focused), and behavioral reports of probabilities were kept to a minimum. The design had no overt rewards to minimize confounds related to valuation processes. Finally, probability is a latent parameter of this task (i.e., it is not observable in stimulus space), which minimizes confounds related to sensory processes.

To anticipate our result, we provide evidence for the non-monotonicity of the code of probability, and we strengthen this conclusion by comparing it to another quantity, confidence, whose code is monotonic. There is evidence from previous studies that the confidence that accompanies a probability estimate in a learning task correlates (linearly) with neural activity, particularly in regions of the fronto-parietal network[11,31,32]. The comparison between probability and confidence is not confounded by between-subject differences or differences in perceptual space since both are derived from the very same sequence of stimuli.

## Results

### Task probing the representation of probabilities

We subjected twenty-six human participants to a probability learning task and simultaneously measured their brain activity using ultra-high field (7T) fMRI to examine their neural representation of probabilities, and of the confidence associated with their probability estimates. After one training session outside the scanner, participants performed four sessions of the task in the MRI scanner. In each session, participants observe a sequence of 420 stimuli appearing one by one, each with a binary value (A or B) sampled from a hidden generative probability p(A) (Fig. 1). This hidden probability undergoes abrupt change points at random unpredictable times, which were not signaled to the participants. The participant's goal is to estimate the current value of the hidden probability throughout the sequence. To perform the task correctly, participants have to frequently update their estimate as new observations are made. This experimental protocol allows us to efficiently probe the neural representation of probabilities and confidence as it induces frequent variations in these quantities that can be compared with variations in measured neural activity.

We used an occasional report paradigm to minimize confounds related to reporting and motor processes in the fMRI signals. We asked participants to perform their estimation covertly on most trials, on which we focused for the fMRI analysis, and only occasionally requested behavioral reports. The reports occurred during dedicated periods (on average every 22 stimuli), during which the participant had to select a range for their current probability estimate and associate a level of confidence with that estimate. To further minimize confounds related to reporting and motor preparation, the probability and confidence questions were counterbalanced across participants. We found a strong correspondence between the estimates reported by the participants and those of a normative model of the task, see Methods for details on the model (Fig. 1 B; Pearson $r = 0.79$ mean $\pm$ 0.02 s.e.m, $t_{25} = 31.8$, $p = 9.9 \times 10^{-22}$, 95%-CI = [0.74, 0.84], Cohen's $d = 6.4$ for probability, and $r = 0.16 \pm 0.03$, $t_{25} = 5.9$, $p = 3.6 \times 10^{-6}$, 95%-CI = [0.10, 0.21], Cohen's $d = 1.2$ for confidence). The distributions of probability and confidence reports were also largely similar between participants and the normative model (see Supplementary Fig. 1). In another recent behavioral study with a probability report on every trial, we found that this strong correspondence with the normative model was consistently observed at the trial-by-trial level[33]. The normative model being a reasonable model of the participant's reports, and in addition, a principled model to study learning, we used the estimates of the normative model as a proxy for those of the participants during the

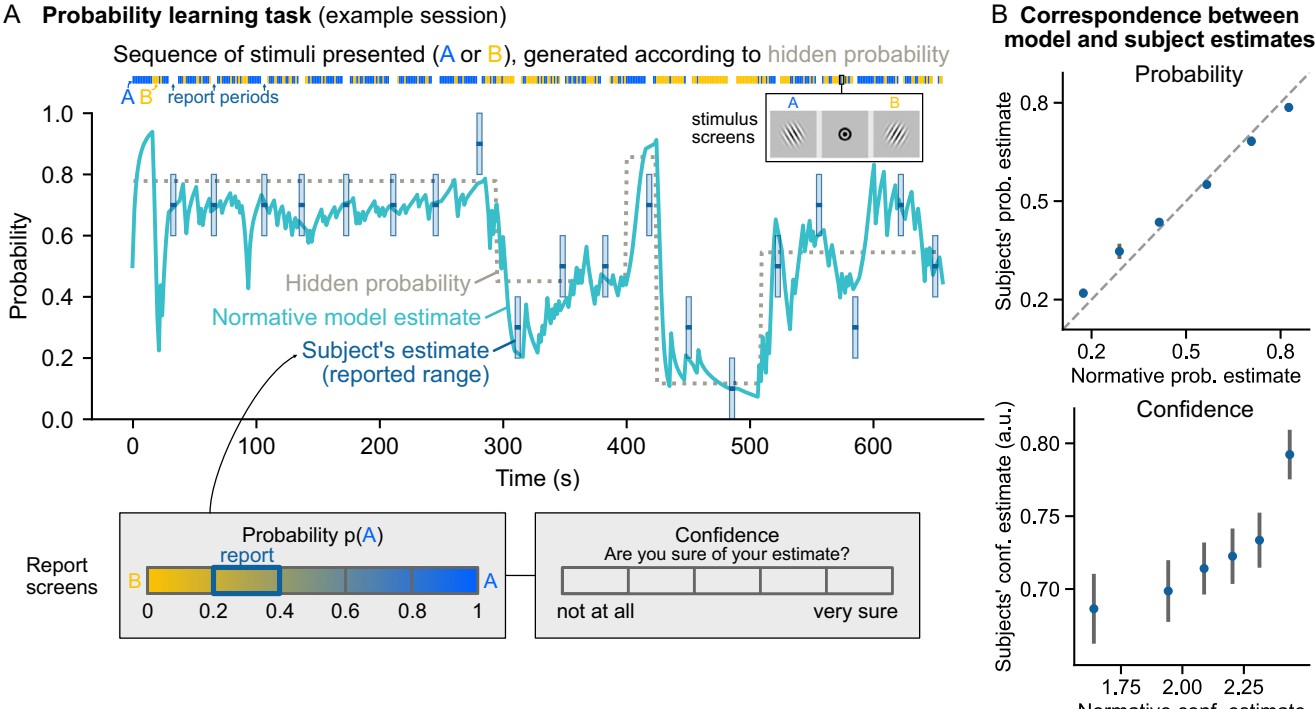

**Fig. 1 | Task and variables of interest. A** Task session. Visual stimuli of two types, **A**, **B** generated according to a hidden probability p(A), are presented successively to the participant. The inset shows a succession of stimulus **A** then stimulus **B** (separated by a fixation dot). During the sequence, the hidden probability changes at random and unpredictable times. Throughout the session, the participant must estimate the current value of the hidden probability. Occasionally between two stimuli, a report period occurs during which the participant selects a range for their current probability estimate and a confidence level associated with their estimate. The normative model of the task is used to obtain the trial-by-trial estimates that the participant should neurally represent. The values of these different variables are displayed above for an example session. **B** Behavioral results: The estimates reported by the participants and the normative estimates are very close for probability (top), and quite correlated for confidence (bottom). Participants' estimates were binned into six equal quantiles of normative estimate, averaged at the participant level and then at the group level. Participants' confidence was recorded from 0 to 1; normative confidence is in log precision units (hence the different scales). Dots and error bars show group-level mean ± s.e.m ($n = 26$ participants). Dashed diagonal is the identity line.

long periods of covert estimation, and related them trial-by-trial with ongoing neural activity.

## Encoding models and evaluation procedure

To relate probability and confidence estimates to neural activity, we constructed models that predict neural activity as a function of the estimate of interest (probability or confidence), which we refer to as encoding models. We defined two classes of models: the class of linear encoding models, which is the one classically used in fMRI studies and which assumes a linear (and by extension, monotonic) relationship between the variable of interest and neural activity, and the class of versatile encoding models, which does not make such an assumption. Instead, the versatile class captures all kinds of relationships, which include but are not restricted to monotonic relationships, by leveraging approximation theory.

In more detail, the versatile model combines two sets of methods. The first corresponds to the principle from approximation theory that any function, and thus in our case the tuning curve of a voxel coding for probability or confidence, can be approximated as a weighted sum of basis functions[23]. This principle is illustrated in Fig. 2. and in a Python notebook (see online code). The second set of methods is the classic regression approach used in fMRI known as General Linear Model (GLM). Both methods are combined by deriving regressors that enter a GLM estimated in each voxel. Regressors correspond to the activity of each basis function in response to the estimates of probability or confidence across trials, that is convolved by a response function accounting for the haemodynamic delay in fMRI (Fig. 3A).

To evaluate the encoding models, we incorporated them into a larger pipeline from which we obtain predictions at the level of the fMRI signal measured in each voxel (Fig. 3A). The resulting model predicts the fMRI time series in a given session from the probability or confidence estimates, as well as several factors of no-interest that we aim to control for, including: the stimulus onset, the surprise elicited by the stimulus, the entropy associated with the probability estimate (a measure of unpredictability), factors related to the report periods, and motion factors.

We fitted the model parameters and tested the fitted model on independent data using a leave-one-session-out cross-validation procedure (see "Methods"). Cross-validation is useful to test how a model generalizes to new sequences of observations, and to prevent overfitting. During testing, we only kept the part of the model corresponding to the factor of interest (probability or confidence) and calculated the $R^2$ score, which represents the predictive accuracy of the model. Finally, to ensure that a positive score can only be obtained by an encoding of probability and confidence related to the sequence specifically observed by the participant and not merely to the statistical structure of the sequences in general, we calculated a null distribution of $R^2$ scores by performing the same analysis after replacing the true sequence with other sequences generated by the same task process, and standardized the score obtained with the true sequence relative to this null distribution to yield the final score, which we call $z$-$R^2$. This combination of analysis choices makes our test of a non-monotonic code for probability more rigorous.

We validated our approach end-to-end using simulations (Fig. 3B). We simulated an experiment following our protocol under

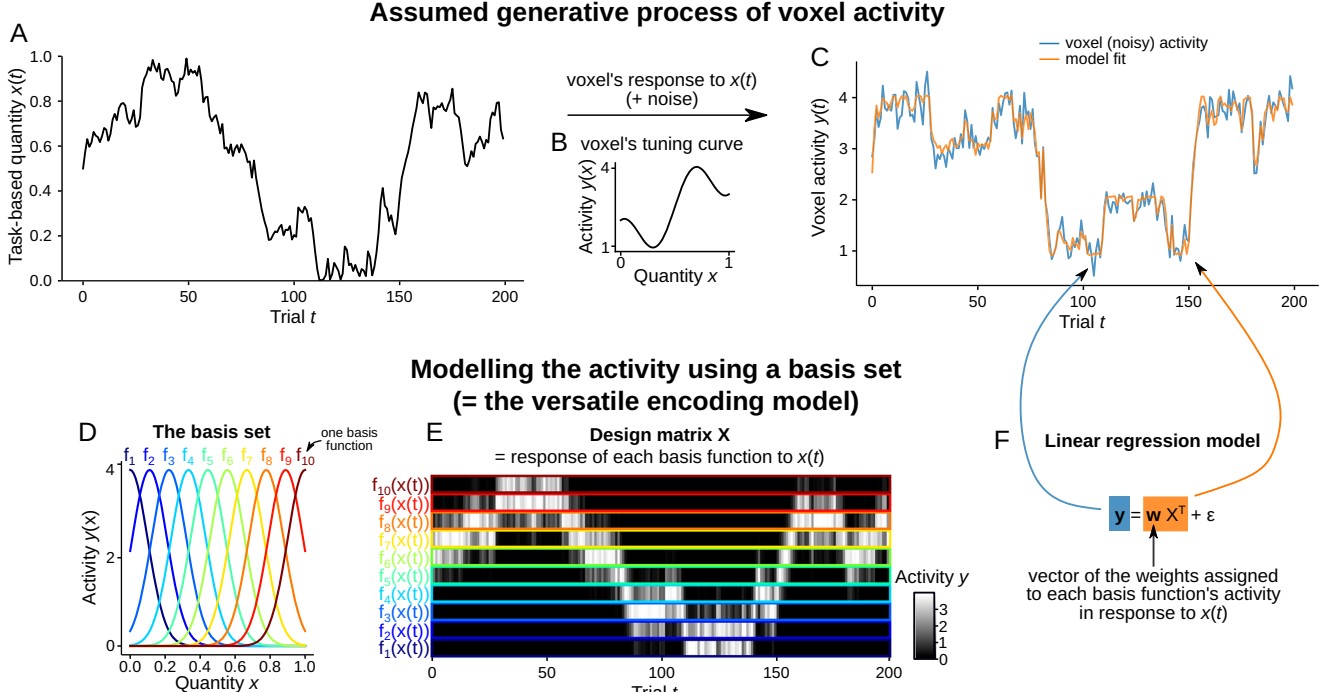

**Fig. 2 | Approximating the activity related to a task-based quantity with a function basis set. A–C** Assumed process for generating voxel activity during the task. **A** Time course of some task-based quantity $x$ across trials; in this study, $x$ would be the estimated probability of the stimulus or the confidence associated with this estimation. **B** We assume that voxels respond to $x$ following a tuning curve, which describes the average activity $y$ of the voxel as a function of $x$. The shape of this tuning curve is unknown to the experimenter, and probably different across voxels. **C** Activity time series of the voxel in response to $x$ (shown in blue), with added noise, given the tuning curve shown in (**B**). **D–F** Description of the approach used to model the voxel activity, without knowing the shape of the tuning curve and making strong assumptions about it. The key idea is that any function can be approximated by a linear combination of basis functions. The versatile encoding model uses this approach. **D** We used a basis set consisting of bell shaped

(Gaussian) functions that tile the range of the quantity $x$. Note that these functions are mere analysis tools, they do not correspond to any assumption about the tuning curves of neurons in the brain. The choice of the basis function does not matter, provided that they tile the range of $x$; for instance, sigmoidal basis functions would work just as well (see Supplementary Fig. 2). **E** The responses of each basis function to the $x$ time course are stacked and displayed as a matrix $X$. Note that when using bell-shaped basis functions, the resulting matrix resembles the time course of $x$ shown in (**A**). The matrix $X$ constitutes the design matrix used for the regression analysis. **F** The voxel activity is modeled as a linear combination of the activity related to $x$ arising from each basis function. The resulting fitted activity time course is shown in **C**, orange curve. The method works for any (unknown) tuning curve, provided that it is sufficiently smooth.

the hypothesis that neural activity noisily encodes one of the two types of estimate (probability or confidence) according to an encoding belonging to one of the two classes (linear or versatile), and by applying our fMRI analysis pipeline to the simulated fMRI signal. This analysis established that when neural activity encodes the same type of estimate as the model, the linear model only explains the signal well when this encoding is linear, while the versatile model explains the signal well no matter whether this encoding is linear or non-monotonic. In other words, a non-monotonic code can only be detected by the versatile model, while a monotonic code can be detected by both the linear model and the versatile model. When neural activity and the model encode a different type of estimate, the model does not explain the signal. In other words, an encoding of probability cannot be mistaken for an encoding of confidence, or vice versa. Several other aspects of these simulations are worth noting. Whatever the code, it should be easier to detect correlates of confidence than of probability (because $z$-$R^2$ is larger on average). Crucially, despite different distributions of confidence and probability during the task (see Supplementary Fig. 1), the ability to correctly detect a non-monotonic code is high for both confidence and probability. In particular, the simulations show that the task and the analysis do not bias the results in favor of a non-monotonic code for probability rather than for confidence, indicating that the empirical results detailed below do not arise from a methodological artifact.

## Neural encoding of probability

The versatile model revealed a significant encoding of probabilities in the prefrontal and parietal cortex (Fig. 4 and Table 1). In contrast, the linear model did not reveal any encoding of probabilities across the cortex, even below the significance threshold (see Fig. 4). The superiority of the versatile encoding model over the linear one is confirmed by the statistical map of the difference in $R^2$ between the two models (Supplementary Fig. 2).

This pattern of results is consistent with those predicted under the hypothesis that the code for probabilities in voxel activity is non-monotonic (Fig. 3B, second row and first two columns of the matrix). Note that these results do not depend much on the specific choice of basis functions (e.g., Gaussian or sigmoidal, see Supplementary Fig. 3).

## Neural encoding of confidence

Contrary to probabilities, for confidence, the linear encoding model significantly explains the measured signal in large regions around the intraparietal and precentral sulci (Fig. 5 and Supplementary Table 1). This is consistent with previous studies on the neural correlates of confidence[11,32]. As expected, the versatile encoding model also captures the signal in the regions explained by the linear model. Interestingly, there was no region captured by the versatile encoding model that was not already captured by the linear encoding model (Fig. 5), indicating that the code for confidence is essentially reducible to a monotonic one throughout the cortex. This indicates that probability

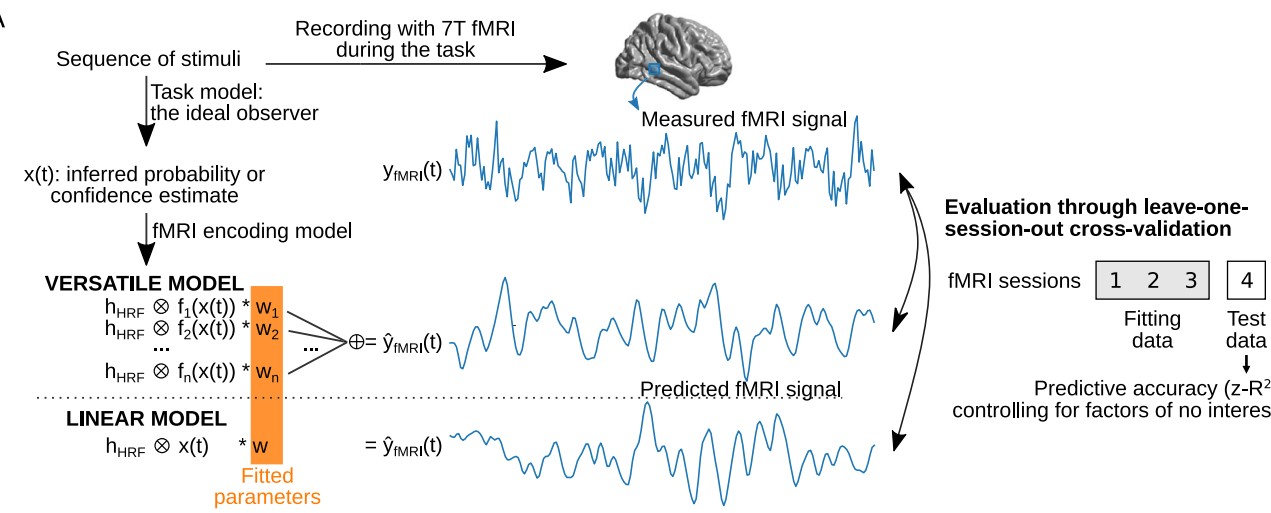

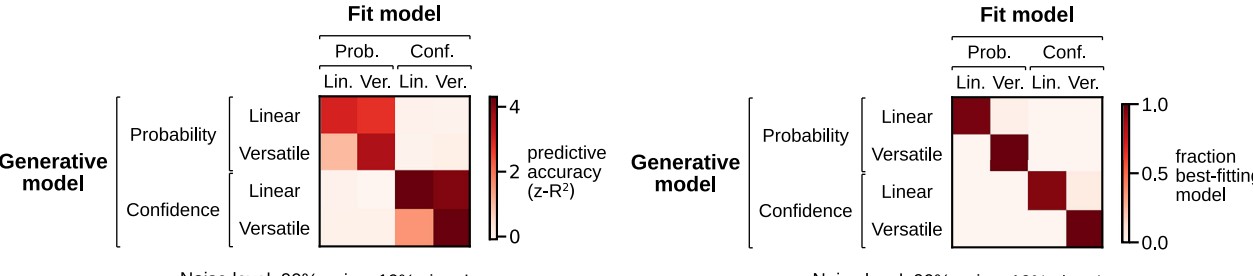

**Fig. 3 | Modeling the encoding of probability or confidence estimates in voxel-wise fMRI signals. A** Schematic of the encoding models and their evaluation against fMRI data. For each session, the sequence of stimuli presented to the participant is given to the task model to obtain $x(t)$, the probability or confidence estimate at time $t$. These estimates across time are then given to the encoding model to predict $y_{fMRI}(t)$, the fMRI signal time series in a voxel. Two classes of encoding models were tested: the linear class, in which fMRI activity in a given voxel is a linear function of the estimate $x(t)$, and the versatile class, in which it is a weighted sum of basis functions ($f_i$) of $x(t)$ that can approximate linear and non-linear functions of $x$ (including non-monotonic ones). Note that the versatile model follows the principle explained in Fig. 2, with the addition here of the convolution (denoted by $\otimes$) with the haemodynamic response function $h_{HRF}$, which is standard in fMRI analysis. To evaluate the models, we used a cross-validation procedure in which three out of four sessions were used to fit the encoding model, and the left-out session was used to measure the predictive accuracy of the model ($z$-$R^2$) after controlling for factors of no interest. **B** Simulation results validating the approach end-to-end. Noisy fMRI data for one experiment were generated assuming a given model of neural activity, and the generated data were used to evaluate each of the encoding models using the procedure described in (**A**). The matrices show the average score (left) and the fraction of simulations where a given model best explained the data (right) across simulated experiments. Linear models explain the data well only when the generative model is linear, whereas versatile models explain the data well for both classes of generative model. Probability and confidence are well separated by the models. The best fit model is almost always the generative one.

and confidence are encoded differently by the brain, which we sought to further characterize next.

**Univariate characterization of the code**

To characterize the code, we reconstructed the tuning curves measured at the vertex level (a vertex is the equivalent of a voxel when working on the cortical surface). These curves were obtained by calculating the sum of the basis functions weighted by the fitted weights of the versatile model (Fig. 6A, see also the notebook explaining the method in the online code). For each participant, we focused on a set of vertices where the predictive accuracy of the versatile model was large enough to ensure a reliable characterization of the tuning curves, which we verified by estimating the tuning curves independently on two halves of the data (Pearson correlation of the estimated weights on the two halves for the vertices of interest: $0.55 \pm 0.03$ and $0.51 \pm 0.04$ for probability and confidence, respectively—see "Methods"; Definition of vertices of interest for characterization). See examples of tuning curves estimated on the whole and on two halves of the data in Fig. 6B.

We used three characteristic measures to describe and compare the tuning curves for probability and confidence (Fig. 6C and Supplementary Figs. 4 and 5). For all three measures, the neural encoding of probability and confidence differed significantly. The first measure looked at the location of the maximum of the tuning curve (i.e., the probability or confidence value that maximizes neural activity). This maximum is expected to be close to one of the two extremes in the case of a monotonic code. Compared to confidence, tuning curves for probability were more often maximized at non-extreme values (Fig. 6C, proportion: $84 \pm 6\%$ for probability vs. $42 \pm 6\%$ for confidence; two-tailed independent t-test of the difference in proportion: $t_{38} = 4.7$, $p = 3.3 \times 10^{-5}$, 95%-CI = [24%, 59%], Cohen's $d = 1.5$). The second measure assessed the degree of non-monotonicity of the tuning curves, quantified by a continuous index between 0 and 1 (see "Methods" and illustration in Fig. 6C). According to this index, tuning curves were more non-monotonic for probability than for confidence (Fig. 6C, $0.61 \pm 0.04$ for probability vs. $0.36 \pm 0.04$ for confidence; two-tailed independent t-test of the difference in non-monotonicity index:

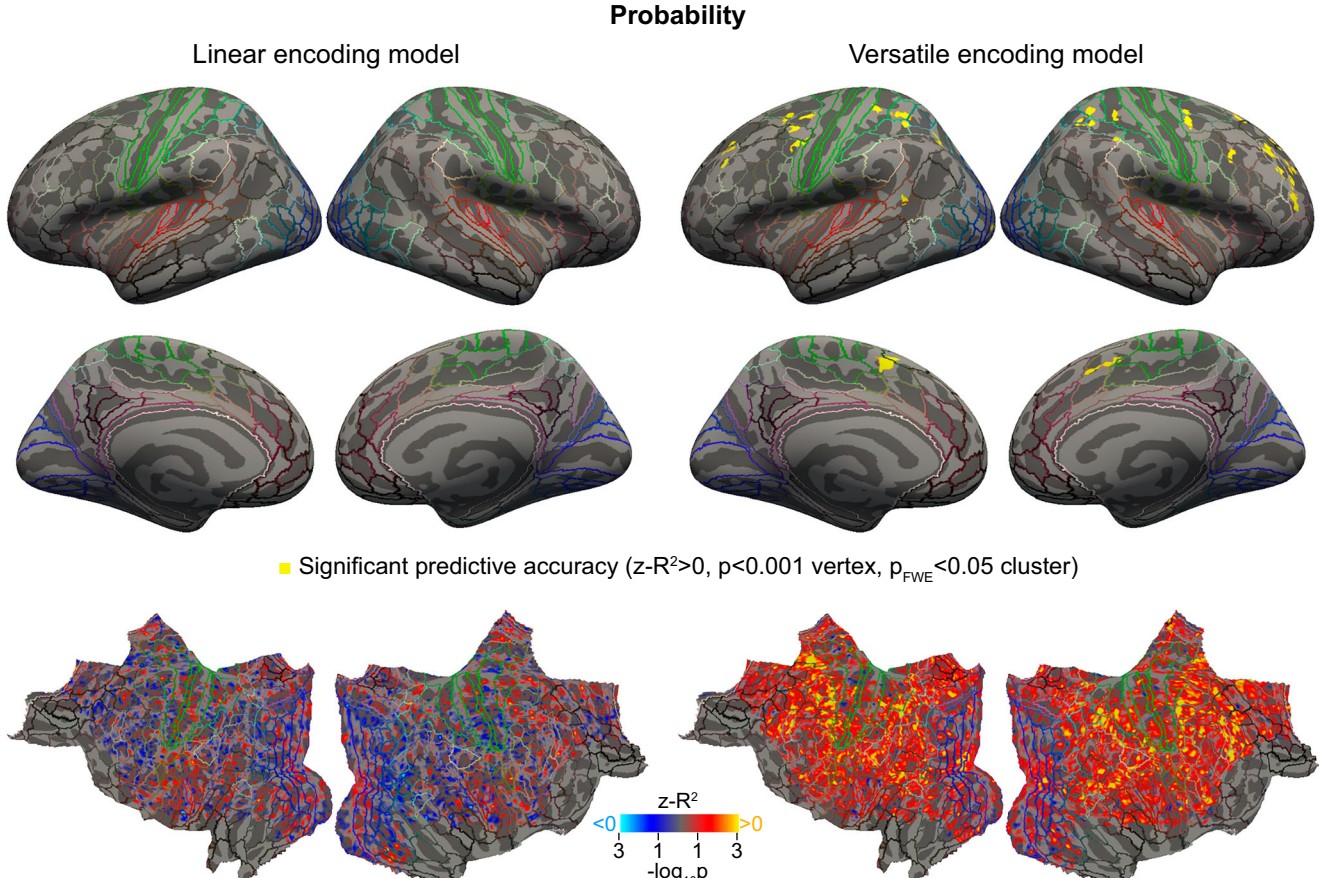

**Fig. 4 | A code for probabilities revealed by whole-cortex analysis in prefrontal and parietal cortex, that only the versatile model is able to explain.** Cortical maps above show the significance of the cross-validated performance of the linear and versatile encoding models (vertex-wise $p$ values obtained from a one-sample t test against zero). Top maps show the significant regions after thresholding at $p < 0.001$ at the vertex level and correcting for multiple comparisons using cluster correction at a family-wise error rate of $p_{FWE} < 0.05$. Bottom maps show the $p$-values without thresholding or correction on a flattened view of the cortex. $P$-values correspond to the group-level significance of $z$-$R^2$ scores obtained across participants (cold colors for negative scores, hot for positive scores). The colored lines indicate the HCP-MMP1.0 parcellation[34].

$t_{38} = 4.6$, $p = 4.1 \times 10^{-5}$, 95%-CI = [0.14, 0.36], Cohen's $d = 1.5$). The third measure looked at the degree of nonlinearity of the tuning curve, quantified as for non-monotonicity by a continuous index between 0 and 1 measuring the deviation of the tuning curve from linearity (see "Methods" and illustration in Fig. 6C). This measure showed that tuning curves were more nonlinear for probability than for confidence (Fig. 6C, 0.87 ± 0.03 for probability vs. 0.45 ± 0.04 for confidence; two-tailed independent t-test of the difference in nonlinearity index: $t_{38} = 7.2$, $p = 1.3 \times 10^{-8}$, 95%-CI = [0.30, 0.53], Cohen's $d = 2.3$).

**Multivariate characterization of the code**

So far, our characterization has been univariate, focusing on one voxel at a time. We now turn to multivariate analyses, which investigate how a population of voxels, collectively, codes for probability or confidence. As explained below, monotonic and non-monotonic codes should result in different geometries of population activity. Characterizing this geometry should therefore provide another test of the hypothesis that the code for probability is highly non-monotonic. We present below two kinds of multivariate analysis. First, using a multivariate decoding approach, we tested whether patterns of voxel responses are informative about probability (and confidence), which would be expected if different voxels are tuned to different ranges of probability (and confidence), as indicated by the above encoding analysis. Second, we characterized the geometric properties of the patterns of voxel responses arising from a non-monotonic vs.

monotonic code, and compared them to the geometry of the fMRI patterns that code for probability and confidence.

We measured the extent to which probability (and for comparison, confidence) could be decoded in different brain regions by dividing the cortex following the parcellation by Glasser et al.[34] and pooling homolog regions in both hemispheres, resulting in 180 regions. We based our decoding approach on the versatile encoding model presented above. The versatile encoding model quantifies in each voxel the weights of basis functions of probability. When the basis functions are equally spaced narrow Gaussian functions as here, then the set of estimated weights characterizes the voxel responses to different bins (i.e., narrow ranges) of probability (to illustrate, see Fig. 2A, E). We tested whether it is possible to identify (i.e., decode) the probability bin that elicited a given pattern of voxel responses. We used 5 probability bins instead of 10 as in the encoding approach presented above to make the estimates of weights more reliable, and thus decoding easier. We adopted the same approach to decode confidence levels.

We trained and tested the decoder on different data sets, using leave-one-session-out cross validation at the participant-level[35]. First, we reduced dimensionality in each region by selecting the 100 voxels that are the most informative about probabilities (using the $z$-$R^2$ metric introduced above, but estimated using the training sessions only). Then we estimated the patterns of voxel responses for different probability bins, for the test session on the one hand and the three

**Table 1 | Significant clusters explained by the versatile encoding model for probability**

| Size (mm²) | Num. vertices | Hemisphere | Cortical area | Parcel | Peak x | Peak y | Peak z | Peak -log10(p) | Cluster p_FWE |
|---|---|---|---|---|---|---|---|---|---|
| 160.4 | 232 | Right | Dorsolateral_Prefrontal | a9-46v_R | 38.9 | 45.2 | 16 | 6.4097 | 0.0002 |
| 157.4 | 183 | Left | Early_Visual | V2_L | −25.1 | −97.9 | −2.7 | 4.8187 | 0.0002 |
| 139.01 | 236 | Left | Paracentral_Lobular_and_Mid_Cingulate | SCEF_L | −9.5 | 5.3 | 55.9 | 4.8964 | 0.0002 |
| 108.66 | 227 | Right | Premotor | FEF_R | 42.2 | −5.8 | 50.7 | 6.6067 | 0.0002 |
| 106.8 | 274 | Right | Superior_Parietal | LIPv_R | 27.7 | −52.8 | 47.4 | 5.211 | 0.0002 |
| 101.02 | 198 | Left | Superior_Parietal | 7PC_L | −37.4 | −50.2 | 55.6 | 4.5741 | 0.0002 |
| 100.41 | 202 | Right | Dorsolateral_Prefrontal | 46_R | 27.3 | 29.6 | 36.2 | 4.0095 | 0.0002 |
| 98.83 | 176 | Left | Premotor | 6a_L | −29.1 | 0.6 | 48.6 | 5.3333 | 0.0002 |
| 80.1 | 169 | Right | Superior_Parietal | 7PC_R | 35.1 | −49.1 | 60.5 | 4.1947 | 0.0006 |
| 71.32 | 161 | Left | Inferior_Parietal | IP1_L | −31.3 | −64.2 | 40.1 | 5.3758 | 0.0014 |
| 70.44 | 127 | Left | Premotor | FEF_L | −37.3 | −7.6 | 44.5 | 4.6641 | 0.0014 |
| 69.62 | 143 | Right | Paracentral_Lobular_and_Mid_Cingulate | 24dd_R | 9.5 | 1.1 | 53.9 | 4.8056 | 0.0008 |
| 69.09 | 132 | Left | Dorsolateral_Prefrontal | 8Ad_L | −25.1 | 24.6 | 33.7 | 5.6881 | 0.0014 |
| 67.68 | 105 | Right | Dorsolateral_Prefrontal | s6-8_R | 21.7 | 20 | 53.1 | 4.3053 | 0.0008 |
| 64.72 | 136 | Left | Premotor | PEF_L | -51.2 | -2.6 | 40.4 | 4.6327 | 0.002 |
| 55.09 | 83 | Right | Dorsolateral_Prefrontal | 9p_R | 19.4 | 38.9 | 37 | 5.5244 | 0.00599 |
| 53.88 | 95 | Right | Dorsolateral_Prefrontal | 46_R | 35.1 | 33.9 | 27.9 | 4.6328 | 0.00719 |
| 51.51 | 93 | Right | Dorsolateral_Prefrontal | 9-46 d_R | 30.3 | 38.8 | 20.9 | 4.7414 | 0.00918 |
| 49.07 | 57 | Left | Primary_Visual | V1_L | −17.4 | −100.9 | 0.2 | 4.189 | 0.01296 |
| 48.56 | 70 | Left | Dorsolateral_Prefrontal | 46_L | −39.2 | 31.5 | 29.3 | 4.2904 | 0.01355 |
| 48.14 | 103 | Left | Somatosensory_and_Motor | 2_L | −36.9 | −35.2 | 56.7 | 4.5609 | 0.01415 |
| 45.65 | 96 | Right | Dorsolateral_Prefrontal | 8C_R | 33.9 | 10.8 | 33.4 | 4.4846 | 0.0203 |
| 44.86 | 69 | Right | Dorsolateral_Prefrontal | 9-46d_R | 25.4 | 38.8 | 32.1 | 4.2132 | 0.02247 |
| 43.94 | 88 | Left | Premotor | 55b_L | −45.1 | 1.5 | 46.8 | 3.7926 | 0.02366 |
| 43.3 | 56 | Right | Dorsolateral_Prefrontal | 9-46d_R | 25.2 | 45.1 | 29.1 | 4.2352 | 0.02702 |
| 43.14 | 84 | Right | Inferior_Parietal | IP2_R | 50.1 | −37 | 45.4 | 5.2945 | 0.02721 |
| 41.56 | 123 | Right | Somatosensory_and_Motor | 2_R | 31.5 | −35.1 | 46.2 | 3.9812 | 0.03292 |
| 41.06 | 91 | Left | Superior_Parietal | AIP_L | −31.1 | −48.1 | 44.3 | 4.2868 | 0.0345 |
| 38.14 | 72 | Left | Temporo-Parieto-Occipital_Junction | STV_L | −53.8 | −46.6 | 11.7 | 4.2271 | 0.04879 |

The names of the cortical areas and parcels refer to the HCP-MMP1.0 atlas[34]. Peak x, y, z are MNI coordinates. Statistical analysis as in Fig. 4.

training sessions together on the other hand. Last, we decoded the probability bin corresponding to a response pattern in the test session by identifying the probability bin eliciting the most similar response pattern in the training session. We computed the decoding accuracy for each brain region and tested for statistical significance against chance-level accuracy at the group level (see Fig. 6 and "Methods").

Decoding accuracy was generally larger for confidence than probability (Fig. 7, Supplementary Tables 2 and 3), which is expected given that the versatile encoding model accounted for voxel responses in more regions based on confidence than on probability (Figs. 5 vs. 6). For probability, we found 6 regions with FDR-significant decoding accuracy. They comprised the dorsolateral prefrontal cortex and the intraparietal cortex, which had been identified with the encoding approach. For confidence, more regions exhibited an FDR-significant decoding accuracy, notably in parietal and prefrontal cortex, similar to the regions identified with the encoding approach.

Decoding accuracy indicates that patterns of voxel activity are informative about probability or confidence, but it does not characterize the type of code being used. We examined more closely the patterns of voxel responses to test for the existence of different codes for probability and confidence. More precisely, we examined the matrices that quantify the dissimilarity of patterns of voxel responses between bins of probability (and similarly, bins of confidence) that served as a basis for decoding. These matrices are called representational dissimilarity matrices (RDM)[36]. Different types of code predict different types of RDM (Supplementary Fig. 6). If the code is highly non-monotonic, then the patterns of voxel responses should be maximally similar when representing the same bin, and equally dissimilar between a given bin and any other bin, resulting in an identity RDM (Fig. 8A). In contrast, if the code is highly linear (or more generally monotonic), then the patterns of voxel responses should be more similar for bins that are closer to one another, resulting in a graded RDM. To determine which code best accounted for the empirical RDMs, we regressed the RDMs for probability (and confidence) obtained in each region onto the identity RDM and graded RDM arising from highly non-monotonic and monotonic codes, respectively.

We first focused on the 5 regions exhibiting the most significant decoding accuracy, separately for probability (Fig. 8B) and confidence (Fig. 8C). The identity RDM significantly accounted for the RDMs for probability, while no significant effect of the graded RDM was found, and the difference between the two approached significance in most regions. In contrast, for confidence, the graded RDM significantly accounted for the RDMs, the identity RDM yielded no significant effect, and their difference was significant in most regions.

We then carried out another analysis that covered all regions. We counted the number of regions best explained by the identity or the graded RDM based on the maximum regression coefficient. A majority of regions followed the graded RDM in the case of confidence ($M_{graded} = 0.66$, SEM = 0.0485, $t_{25} = 3.3$, $p = 0.003$, 95%-CI = [0.561, 0.76], Cohen's $d = 0.66$, two-sided t-test against 0.5) whereas a majority of regions followed the identity RDM in the case of probability ($M_{graded} = 0.436$, SEM = 0.0348, $t_{25} = -1.8$, $p = 0.078$, 95%-CI = [0.364, 0.508], Cohen's $d = -0.37$, two-sided t-test against 0.5), and the

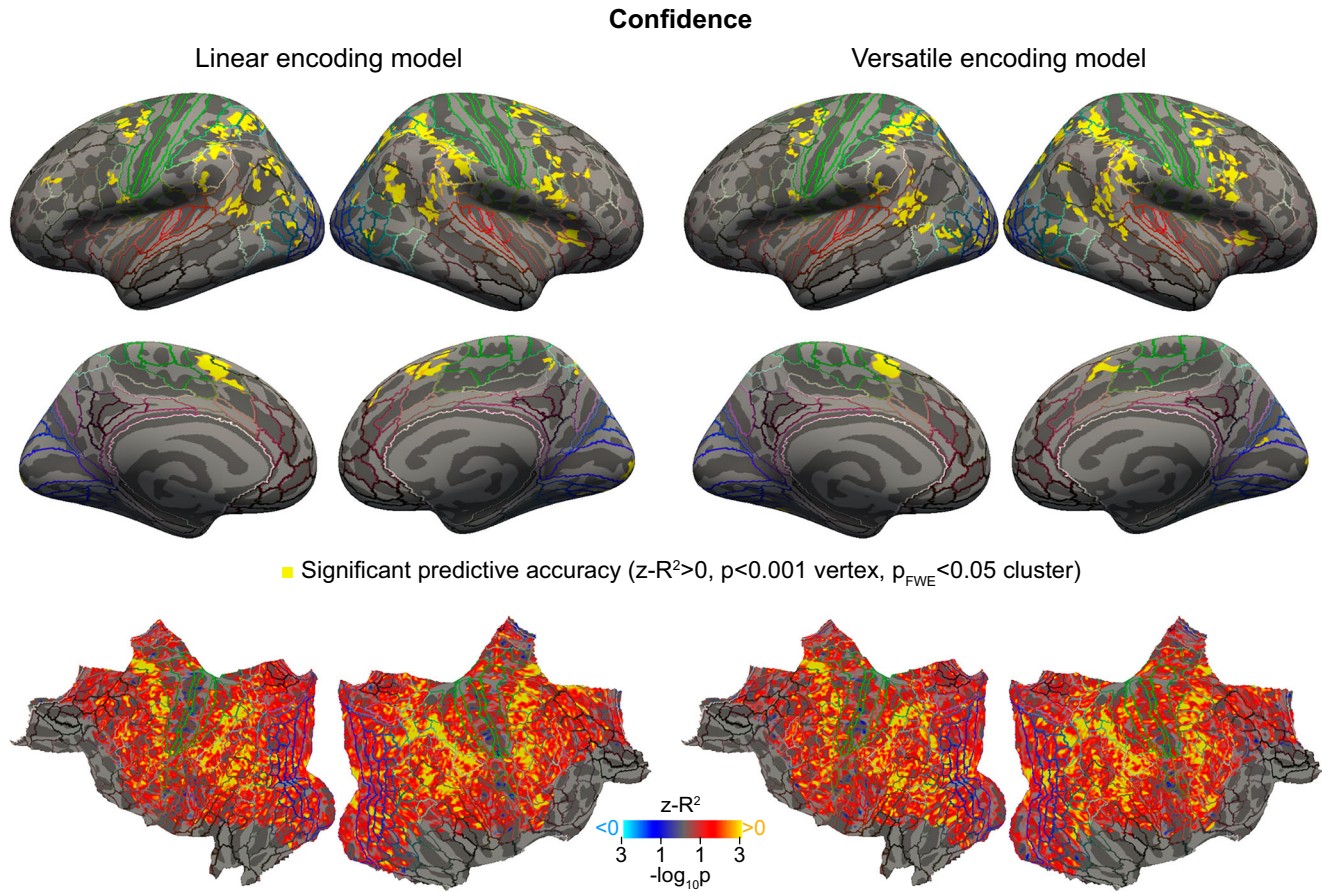

**Fig. 5 | Code for confidence. Both the linear and the versatile models explain the neural encoding of confidence in approximately the same regions of the cortex.** Plotting conventions and statistical analysis as in Fig. 4.

difference between confidence and probability was significant (paired difference: 0.224, SEM = 0.0663, $t_{25}$ = 3.4, $p$ = 0.002, 95%-CI = [0.0878, 0.361], Cohen's $d$ = 0.68, two-sided t-test). Overall, the results of the multivariate characterization taken together with those of the univariate characterization indicate that the codes for probability and confidence are different, being respectively highly non-monotonic and monotonic.

## Discussion

We used a task in which participants estimated the latent probability of an affectively neutral event occurring in a sequence. Participants accurately tracked this latent probability, as revealed by the comparison of their reports with the normative model. We identified a representation of this latent probability in the fMRI signals recorded outside of the report periods, particularly in the dorsolateral prefrontal cortex and intraparietal cortex. We characterized this representation as based on a non-monotonic code. In contrast, the representation of the confidence that accompanied the probability estimate was based on a monotonic code. Detailed analysis revealed that the vast majority of fMRI voxel tuning curves for probability were non-monotonic, with one or more local extrema. In contrast, the tuning curves for confidence were essentially monotonic. In addition, a multivariate analysis of the patterns of voxel responses revealed again that probability and confidence were represented based on and a monotonic code, respectively, at the voxel population level.

Our results relied on the use of a versatile encoding model capable of accommodating tuning curves of almost any shape. To this end, we combined the universal function approximation properties of basis sets[23] with the use of a GLM for fMRI (Figs. 2 and 3A); the resulting

model corresponds technically to a *linearizing encoding model*[24,25]. This method consists of applying basis functions to a quantity of interest to obtain "features" (our regressors) and then modeling the fMRI signal in each voxel as a linear combination of these features, following the GLM approach that is massively used in fMRI[27] The small number of features (~10) and the one-dimensional nature of the quantity of interest (probability or confidence) facilitate the interpretation of the model in terms of tuning curves. This approach is related to other methods that can accommodate non-monotonic tuning curves, such as population receptive field (pRF) mapping[26,37]. The key difference is that pRF methods assume a specific form of non-monotonicity, typically bell-shaped tuning curves, corresponding to the idea that a voxel is selective for a range of values. In contrast, our method can accommodate tuning curves exhibiting a selectivity for multiple value ranges. Multiple peaks in tuning curves at the voxel level may result from averaging different populations of neurons having single peaks. There may also be multiple peaks at the single neuron level, as reported for neurons sensitive to numbers and as predicted by some models of the number sense[21]. We also note that grid codes, which have been reported in abstract (nonspatial) domains, also posit the existence of multi-peak neurons[38]. Irrespective of the form of the non-monotonicity, our finding that the neural representation of probability is retrospectively explains null findings in previous studies that used methods assuming a linear code[11–13].

Our results raise the puzzling question of why some quantities are encoded with monotonic codes, such as confidence here, or reward[17,39], salience[19], surprise[10,32], prediction error[7], evidence accumulation[3,40] and some other quantities are encoded with non-monotonic codes, such as probability here, or orientation of visual

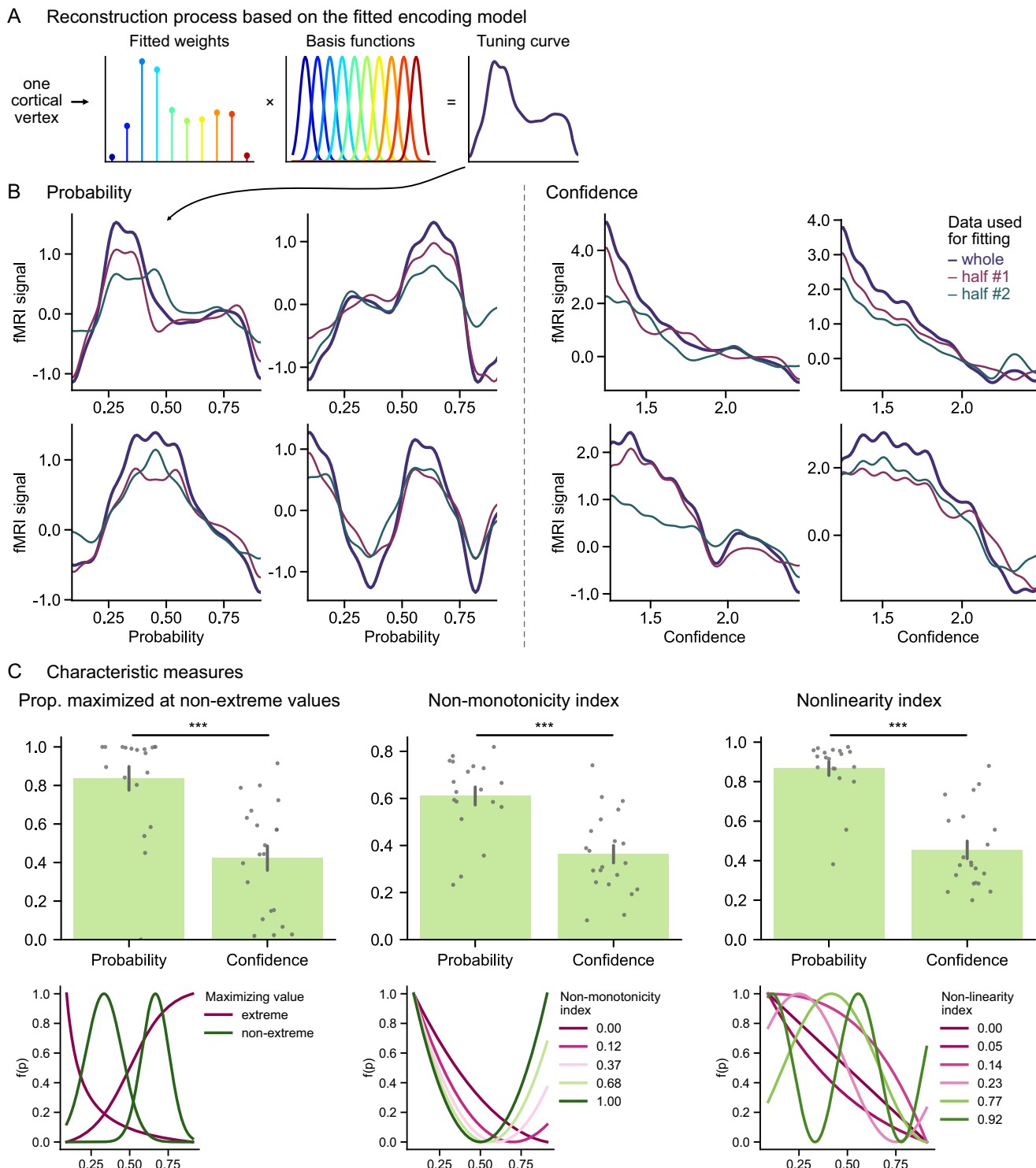

**Fig. 6 | Characterization of the neural encoding of probability and confidence.**
**A** Schematic of the reconstruction of the tuning curve at the level of a cortical vertex. The weights of the versatile encoding model fitted on the vertex data are used to calculate the tuning curve function, which is equal to the weighted sum of the basis functions. **B** Tuning curves obtained for probability (left) and confidence (right) for example vertices and participants (out of approximately 30,000 and 120,000 examples, respectively). Within each panel, multiple curves correspond to multiple estimates of the tuning curve for a same vertex: in purple, the curve estimated with the weights fitted on the whole data, red and green, estimated on two independent halves of the data, illustrating test-retest reliability (which is

ensured, to some degree, by the selection method). **C** Quantitative description and comparison of the tuning curves for probability and confidence according to three characteristic measures. The tuning curves for probability are more frequently maximized at non-extreme values, have a higher non-monotonicity index, and a higher nonlinearity index than those for confidence (see "Methods" for formal definitions). The plots at the bottom show simulated examples of tuning curves for probability and their characteristic measures. Bar heights and error bars show mean ± s.e.m across participants ($n = 26$). Dots show individual participants. ***: $p < 0.001$, two-tailed t-tests (proportion: $p = 3.3 \times 10^{-5}$; non-monotonicity: $p = 4.1 \times 10^{-5}$; nonlinearity: $p = 1.3 \times 10^{-8}$).

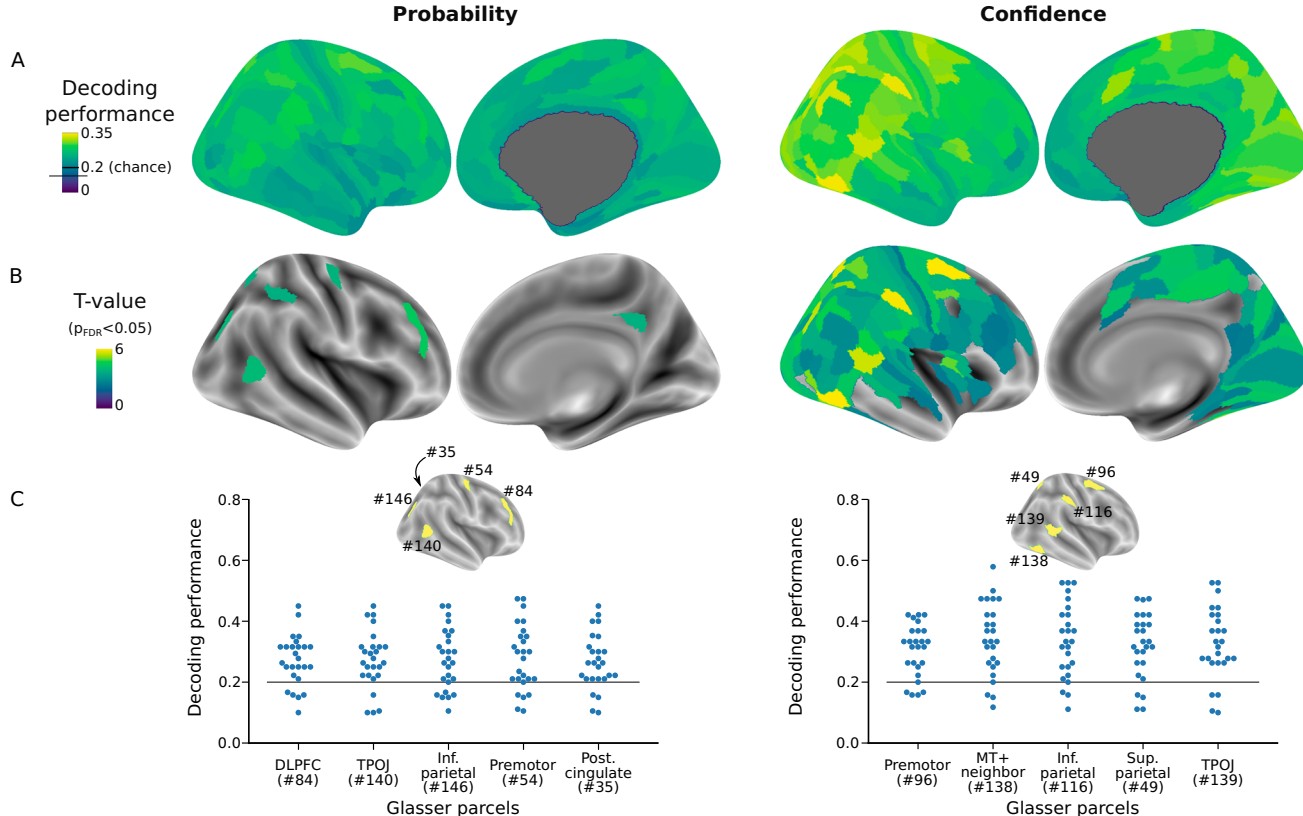

**Fig. 7 | Decoding probability and confidence from the voxel response patterns obtained with the versatile encoding model.** Group-level decoding accuracy for each cortical region (180 bihemispheric parcels from the HCP-MMP1.0 atlas, rendered on the right hemisphere for illustration purpose). **A** Mean accuracy across participants (chance level is one out of five bins, i.e., 0.2). **B** *T*-values of a two-tailed t-test for accuracy different from chance level. Only regions statistically significant after FDR-correction ($p < 0.05$) for multiple comparisons across the 180 regions are displayed (also see Supplementary Tables 2 and 3). **C** Decoding performance of each participant in the five regions with top decoding significance. The decoding method is based on the similarity (correlation distance) of voxel response patterns estimated by the encoding model in a training set and a test set (see "Methods").

object[14], angle in arm reaching[41], numerosity[16]. This question is beyond the scope of our study, but we mention some speculations. Some scalar quantities lie on an axis whose direction is relevant to the regulation of behavior and brain processes. For instance, humans and other animals generally seek more, not less, rewards[6]. A monotonic code with increasing activity levels for larger rewards may, in downstream circuits, facilitate the invigoration of behavior to obtain larger rewards[42], and the comparison of different reward levels (which, in a monotonic code, simply amounts to comparing activity levels). A similar argument applies to salience, surprise, accumulated evidence, and lack of confidence. Higher values of these quantities usually enhance other processing: more salient and surprising events elicit stronger orienting responses[43], lower confidence about a learned estimate increases the learning rate[31,33]. In contrast, in our task, the probability of occurrence of a right- vs. left-tilted Gabor patch has no valence and it is not immediately relevant to behavior. Interestingly, when a set of tuning curves of neurons or voxels is sufficiently broad and diverse, the average activity level of the corresponding population tends to be the same for different values of the encoded quantity (a property known as invariance). Assuming that more neural activity is costly[44], this invariance property implies that the same energy budget is expended to represent any probability, in particular for low (close to 0) and large (close to 1) probabilities. In contrast, encoding reward, salience, surprise, or lack of confidence with increasing (linear) levels of activity appropriately expends more energy on larger, behaviorally relevant values.

Here, we found a neural representation of probabilities predominantly in the dorsolateral prefrontal cortex, the precentral sulcus and the parietal region. The dorsolateral prefrontal and intraparietal cortices have been reported to host a general coding system for magnitudes of different types, from numbers of perceived objects to internally generated numerical quantities and proportions in humans and monkeys[16,45,46]. Our results suggest that this general coding system may also encode probability. In our results, the dorsolateral portion of the prefrontal cortex appeared to encode only probability, whereas the precentral sulcus and the parietal regions encoded both confidence (with a monotonic code) and probability (with a nonmonotonic code). Simulations showed that a monotonic code for confidence and a non-monotonic code for probability can be clearly distinguished from one another; their co-localization is therefore a notable finding. We speculate that if a region is involved in estimating and encoding of probabilities, it should be more active when the estimate is updated more, which typically occurs when confidence is lower[11,31]. Co-localization of a non-monotonic code for probability and a decreasing (linear) code for confidence would then be expected. Future studies should distinguish between updating and representing probabilities, and our results suggest that these processes may differentially involve the dorsolateral prefrontal cortex on the one hand, and the precentral sulcus and parietal region on the other hand. It also remains to be determined whether the neural representation identified here is of probability per se or of computations related to the probability estimation process (e.g., working memory, attention).

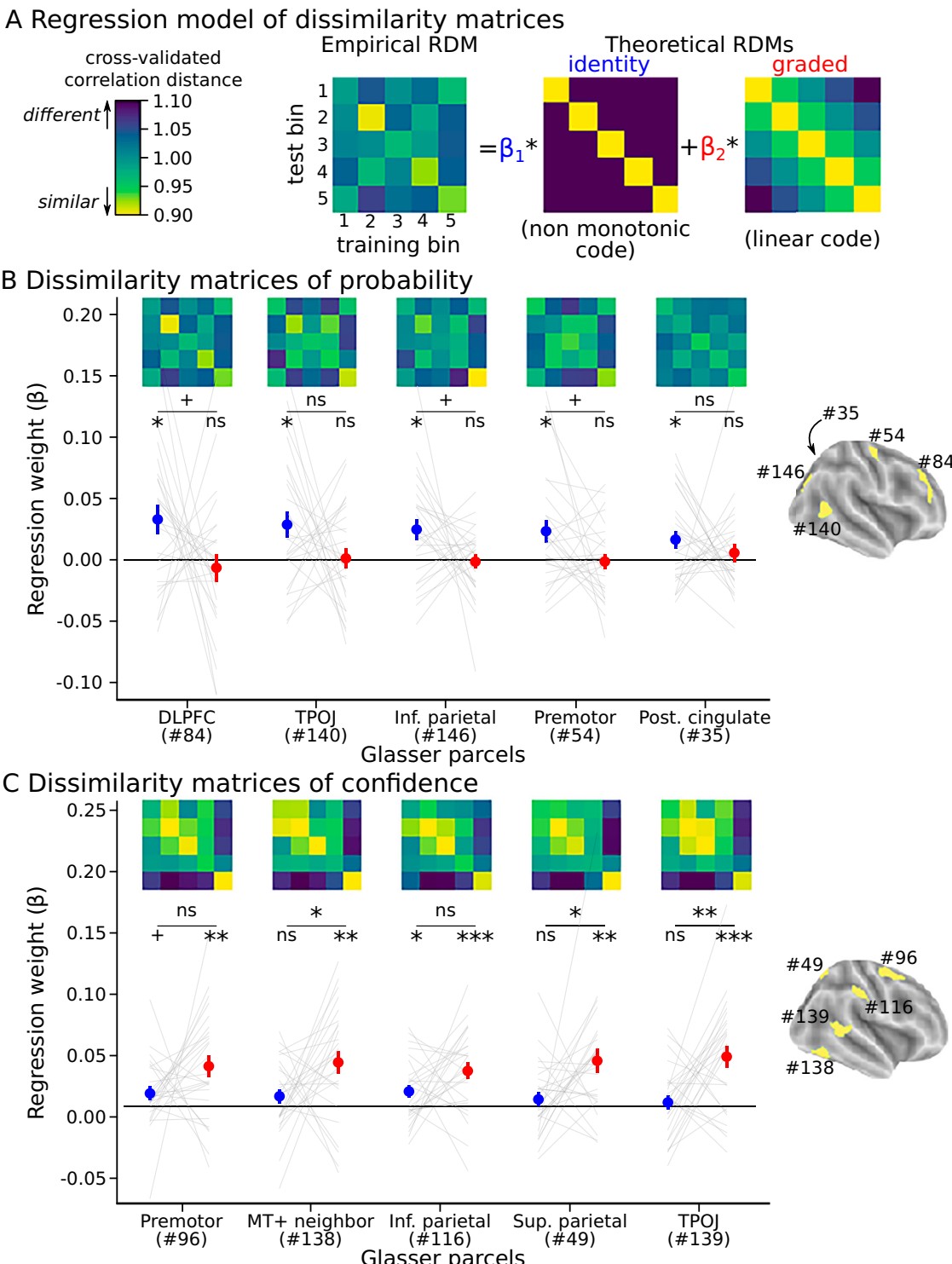

**Fig. 8 | The dissimilarity of voxel response patterns across bins of probability and confidence supports the existence of different types of codes. A** The representational dissimilarity matrices (RDM) that served for the decoding analysis (Fig. 7) were averaged across test sessions and analyzed with a regression analysis at the participant-level. Two theoretical RDMs were considered in the regression: the "identity" RDM, that should result from a highly non-monotonic code, and the "graded" RDM, that should result from a linear (or more generally, monotonic) code. **B** Average RDM of probability levels and regression results in each of the 5 regions with top decoding significance for probability. Each line corresponds to a participant; colored dots and error bars show the average and S.E.M. **C** Same for

confidence. In **B, C** the same color bar (shown in **A**) is used for all RDMs. Each set of regression coefficients was tested at the group level against 0 (two-tailed t-test) and compared with a paired t-test (two-tailed); ns: $p > 0.1$; +: $p < 0.1$; *: $p < 0.05$; **:$p < 0.005$; ***:$p < 0.0005$. The parcel numbers correspond to the Glasser parcellation[34]; 84: Dorsolateral Prefrontal, Area 46; 140: Temporo-Parieto-Occipital Junction, Area 2, 146: Inferior Parietal, Area 0; 54: Premotor, Dorsal Area 6d, 35: Posterior Cingulate, area 31pv; 96: Premotor, Area 6 anterior; 138: MT+ Complex and Neighboring Visual Areas, Area PH; 116: Inferior Parietal, Area PFt; 49: Superior Parietal, Ventral IntraParietal Complex, 139: Temporo-Parieto-Occipital Junction, Area 1.

Our study focused on the neural code at the voxel level (i.e., millimeter scale), as captured by fMRI. While the nature of the neural code at the neural level is beyond the scope of this study, the characterization of tuning curves at the level of voxels has implications at the level of neurons. Since the fMRI activity reflects the average activity of (many) neurons, a non-monotonic tuning curve in a voxel implies that the tuning curves of the neurons it contains must be nonlinear. This is because averaging preserves linearity. In this context, two forms of nonlinearity—monotonic and non-monotonic—must be distinguished. Non-monotonic voxel tuning curves can arise from either non-monotonic single-neuron tuning curves (e.g., as in the versatile model using Gaussian basis functions) or monotonic single-neuron tuning curves (e.g., as in the versatile model using sigmoidal basis functions; Supplementary Fig. 3). However, in the latter case, non-monotonic voxel tuning requires large populations of neighboring neurons with opposing slopes (i.e., both increasing and decreasing) at the millimeter scale. Similarly, constraints also apply to monotonic voxel tuning curves. A monotonic voxel tuning curve can emerge from monotonic single-neuron tuning curves or, under very specific conditions, from non-monotonic single-neuron tuning curves. For instance, in the case of Gaussian single-neuron tuning curves, a steadily increasing voxel tuning curve requires a finely ordered progression of the proportion of neighboring neurons "preferring" higher confidence. Such a precise organization of neuronal preferences at the millimeter scale would be extraordinary. It is therefore more plausible that monotonic voxel tuning curves reflect monotonic tuning at the single-neuron level.

One question, which remains open, is the generality of the neural representation of event probability shown here to other types of probability. Its invariance[4] could be tested with respect to the features of the stimulus (e.g., visual stimuli that differ in shape, or color, rather than orientation) or the sensory modality used (e.g., auditory stimulus[32]). Invariance could be tested by comparing tuning curves estimated from different types of stimuli or decoding probabilities across stimulus types. The task and the cognitive process leading to the probability estimate could also be varied. In terms of task, here we used an occasional report paradigm; finding similar results in the absence of a report of probability and confidence (in a purely passive design, or with an active part that is not a report, e.g., a reaction time task) would further rule out confounding explanations for the present results. In terms of cognitive process, here, the probability results from a statistical learning process operating on a sequence, but it could also result from a reasoning and memory process. Moreover, the probability could refer to other objects than the occurrence of events in the world. One example is estimating the probability that a statement is true, such as "Is Paris bigger than Berlin?"[12]. In addition to tests of domain generality, it would also be useful to study probabilities outside of a learning context in order to examine a broader range of probabilities. Here, we have restricted this range from 0.1 to 0.9, because more extreme probabilities involving rarer events would require much longer sequences to be learned accurately. Investigating more extreme probabilities would also be useful to distinguish a code of the probability itself from its log-odds as suggested by some cognitive models[47], which is not possible in the range investigated here where the two codes are very much linearly related.

We now turn to discuss some limitations of our approach and future directions. First, our ability to differentiate between several ways (which may coexist) of representing probability is partly limited by the fact that we have used binary sequences, which can be described equivalently in terms of p(A) or p(B), since p(A) = 1-p(B). We found evidence that some but not all parts of the cortex exhibit similar activity for values of p(A) that are symmetric with respect to 0.5, e.g., when p(A) = 0.2 and p(A) = 0.8 (see Supplementary Fig. 7). In the parietal and precentral regions, an entropy encoding model (which encodes probability symmetrically) could explain neural activity, as

did the probability encoding model. So it seems that some of the neural correlates of probability we identified may actually be neural correlates of entropy. However, in the dorsolateral prefrontal cortex, the probability encoding model better explained the neural activity. Additionally, the tuning curves for probability obtained in regions identified as encoding probability exhibited less symmetry compared to those obtained in regions encoding entropy (see Supplementary Fig. 8). This indicates that, at least in some regions, the code for probability that we identified cannot be reduced to entropy. Where symmetry was observed, several explanations are possible. The representation of the event probability could switch between p(A) and p(B), or there could be a similar proportion of neurons coding for p(A) or p(B) within a voxel. Another possibility is that some representations of probability may focus on the probability of the most likely stimulus (the one with $p > 0.5$) and the identity of this stimulus (these two pieces of information are sufficient to reconstruct p(A) and p(B) in the binary case). The use of more than 2 items in a sequence would be useful to adjudicate between these different possibilities.

Our results do not address the existence of the topographical organization of the neural representation of probability. In the case of numerosity, this representation has been shown to be spatially organized at millimeter resolution, with smooth changes in the range of the numerosity "preferred" by a voxel across the cortical surface[46]. Our versatile encoding model often identified multiple peaks in probability tuning curves (in 42% of tuning curves on average across participants, s.e.m. 7%); in contrast, the pRF method assumes a bell-shaped tuning curve, with fewer parameters than the versatile encoding model, thereby providing a more robust estimates of a voxel's "preferred" value.

In addition, the timing of the task and the poor temporal resolution of fMRI precluded the use of trial-level decoding, which was done at the session level here. Future studies could explore a decoding of probability across trials, perhaps benefiting from the use of better time-resolved recordings such as electrophysiology. Single-trial level decoding would be useful to test for the behavioral relevance of the neural representation of probability identified in this study. For example, one could test whether the variance and bias of participants' reports relative to the normative estimate are reflected in the decoded probability.

In summary, we have unraveled a neural representation of event probability in the human cortex that is based on a code, and that cannot be detected by simpler methods that assume a monotonic code. The methods we have developed here can be used to search for neural representations in many different types of tasks.

## Methods
### Participants and task
Experimental protocols were approved by the local ethics committee (CPP-100032 and CPP-100055 Ile-de-France) and the informed consent of all 29 participants (15 female, mean age 25.4 ± 1.0 s.e.m.) was obtained before they began the experiment. Three participants were excluded from analysis due to acquisition problems, resulting in an effective total of $N = 26$ participants for analysis.

After receiving task instructions and completing a training session, participants entered the MRI scanner and performed 4 task sessions, during which the scanner recorded functional MRI data. Each session lasted approximately 11 min and consisted of a sequence of 420 stimuli presented one after the other, for 1 s each, with an inter-stimulus interval of 0.3 s (see example in Fig. 1A).

Each stimulus had a binary value, A or B, represented in the task by one of two distinct orientations of a high-contrast grating, easily distinguishable from each other. The values were drawn from a Bernoulli distribution with a hidden generative probability p(A) to be learned, whose value will be denoted $h_t$. The hidden probability $h_t$ followed a stochastic change-point process: at each time step, it could either

remain the same or undergo a change point, with a change-point probability of 1/75 under the constraint of a maximum period of 300 stimuli without change points. The value of $h_t$ was uniformly sampled between 0.1 and 0.9 initially and at each change point, under the constraint that the odds of A changed by a factor of at least four (i.e., $\frac{p_{old}(1-p_{new})}{(1-p_{old})p_{new}}$ is larger than 4 or smaller than 1/4).

Throughout the task, the participants' goal was to estimate the hidden probability $h_t$. The occurrence of unsignaled and unpredictable change points made the estimation all the more challenging. Accurate estimation requires making probabilistic inferences about the value of $h_t$ given the observed stimuli, with knowledge of the generative process of the sequences, as described in the task model below. Participants could make such inferences as they had been briefed about the generative process in an informal way during the instructions phase.

Participants were instructed to continuously estimate the probability during the sequence, and were occasionally asked to report their estimate during dedicated time periods. Isolating the report periods from the periods of stimuli and the estimate updates they elicited allowed us to ensure that neural representation of the factors of interest we found in the fMRI recordings, related to the estimation process, were unlikely to be confounded by the participant's reporting. Report periods occurred on average every 22 stimuli, with a random uniform jitter of ±3 stimuli maximum. Each report period consisted of a response screen where the participant made two choices: a choice of estimate range for $h_t$ among three or five possible ranges (the scale was randomly selected between the three-choice scale and the five-choice scale for each period, except for the first half of participants where the scale was always the five-choice scale), and a choice of confidence level associated with this estimate among five possible levels. The order in which these two choices were made were counterbalanced across participants. These choices were made using a right-handed five-finger button pad. Although the choices were discrete, participants were instructed to internally generate continuous-valued estimates. They were further incited to do so in order to produce correct reports as they did not know in advance which scale, and therefore which choice ranges would be available (there being no trivial mapping between the two scales).

The current study employs a task design inspired by a previous study on probability learning[32]. Some aspects of the design have been improved over that previous study to investigate the neural code of probability. In particular, we changed the statistical properties of the generative process of sequences (a Bernoulli process on each trial, p(A), instead of order-1 transition probabilities, p(A|A) and p(A|B)), used less frequent behavioral reports (every 22 stimuli on average instead of 15), lengthened (420 stimuli instead of 380) and speeded up (SOA: 1.3 s instead of 1.4 s) sequences. The quality of the resulting present dataset is demonstrated by the analysis of behavioral reports and the neural correlates of confidence and surprise published in a recent study[11], which confirmed and extended the results of the original study.

## Task model

The model we used for the task is the normative model, in the sense that it produces, for each trial, the optimal estimate of the hidden probability that the participant could have produced, given the stimuli they have observed so far and their knowledge of the generative process of the task. This estimation problem requires inferring the value of $h_t$ given the stimulus values observed in the past $s_{1:t}$. Since the generative process is probabilistic, the inference problem is also probabilistic (the value of $h_t$ cannot be determined with certainty). Using the rules of probability calculus and in particular Bayes' rule, a posterior distribution on $h_t$ can be calculated, denoted as $p(h_t|s_{1:t})$. In the present case, one solution to calculate algorithmically the posterior distribution is to proceed iteratively on the observations,

initializing $p(h_0)$ to a uniform distribution and computing $p(h_{t+1}|s_{1:t+1})$ based on the value of $s_{t+1}$ and the previously computed $p(h_t|s_{1:t})$. This calculation is done using the following formula.

$$p(h_{t+1} \mid \mathbf{s}_{1:t+1}) \propto p(s_{t+1} \mid h_{t+1}) \int p(h_{t+1} \mid h_t)\, p(h_t \mid \mathbf{s}_{1:t})\, dh_t \quad (1)$$

This formula is derived from the rules of probability calculus by leveraging two conditional independence properties of the present generative process: (1) $s_{t+1}$ is conditionally independent of $s_{1:t}$ given $h_{t+1}$; (2) $h_{t+1}$ is conditionally independent of $s_{1:t}$ given $h_t$. See Supplementary Note 1 for the derivation.

The $\propto$ operator denotes the equality up to a constant factor. This constant is implicitly factored in at computation time by normalizing the right-hand side of equation [1] so that it sums up to 1 over the possible values of $h_{t+1}$, to obtain the left-hand side. The term $p(s_{t+1} \mid h_{t+1})$ is given by the Bernoulli distribution, equal to $h_{t+1}$ or $(1-h_{t+1})$ depending on whether $s_{t+1}$ is A or B. The term $p(h_{t+1} \mid h_t)$ reflects the generative probability that a change point occurs (1/75) or does not occur (74/75), depending on whether $h_{t+1}$ is different from or equal to $h_t$, respectively.

The normative estimate of hidden probability and the associated confidence are both calculated from the posterior distribution. The probability estimate is equal to the mean of the posterior, $E[h_{t+1} \mid s_{1:t}]$. It is optimal in the sense that it minimizes the mean squared error with the true value, and is equal to the posterior probability that the next stimulus value is A (this is formally what participants were asked to report). Confidence was defined as $-log\ SD[h_{t+1}|s_{1:t}]$, the log-precision of the posterior (up to a factor of two). Hereafter, we will use the symbol $x$ to refer to these two types of estimates indiscriminately ($x_t$ representing the estimate for stimulus $s_t$) as the presented encoding models are mostly independent of the specific type of estimate being encoded.

The task model was implemented using Python and the NumPy package (https://numpy.org).

## Encoding models

Encoding models predict the fMRI activity of a voxel as a function of the estimates obtained from the stimulus sequence seen by the participant. We defined a set of 2 × 2 main encoding models, depending on whether the encoded estimates are probability or confidence estimates, and whether the model tuning curve function belongs to the linear or the versatile class (Fig. 3).

We also considered another model, following a hypothesis proposed in the literature, in which the activity was a function of the entire posterior distribution (rather than a moment of the distribution, like the probability and confidence estimates are)[48,49]. That is, in that model, $f_i(x)$ in equation [3] was replaced by its posterior expectation $\int f_i(x)p(x)dx$, where $p$ is the posterior distribution. However, when we tested that model on our fMRI data, it explained the data less well than the simpler models encoding the probability or confidence estimates (see Supplementary Fig. 9). Therefore, we focused on the simpler models.

The probability and confidence estimates are calculated from the sequence as explained in the "Task Model" section above. The model tuning curve function maps the encoded estimate, $x$, to a prediction of neural activity, $\hat{y}$, and is parameterized with weights $w$ that are to be fitted to the data. In the linear class, this function is of the form:

$$\hat{y} = wx \quad (2)$$

In the versatile class, this function is of the form:

$$\hat{y} = \sum_{i=1}^{K} w_i f_i(x) \quad (3)$$

where the $f_i$ are (radial) basis functions that have approximation properties[23]. Here, we used Gaussian basis functions, but we also performed simulations using sigmoid basis functions, and the results were similar whether we used Gaussian or sigmoid functions (Supplementary Fig. 3). The Gaussian basis functions are expressed $f_i(x) = c$ $exp[-(x-\mu_i)^2/(2\sigma^2)]$. The centers of the basis functions $\mu_i$ were distributed to have equal spacing between two consecutive centers, between the lower bound of the interval and the first center, and between the upper bound and the last center (the interval being [0, 1] for probability, and [1.1, 2.6] for confidence). The number of basis functions was $K = 10$, and their dispersion was $\sigma = 0.04$ for probability and $\sigma = 0.06$ for confidence. In principle, larger values of $K$ ensure better approximations of the true tuning curve, but these approximations are also more susceptible to overfitting in noisy data. The value for $K$ was determined through simulations to optimize the $R^2$ scores averaged over a wide range of simulated activity with different $K$ values. The value for $\sigma$ was chosen as a function of $K$ and the domain (which differs for probability and confidence) so that the sum of the basis functions is (approximately) constant, ensuring that all values of the domain are equally considered (translation invariance over the domain).

To transform the predictions of the theoretical models described above into predictions at the level of fMRI activity in a voxel, we convolved the encoding model regressors (that is, the scalar quantities that are multiplied with the weights $w$: $x$ in the linear case and the $f_i(x)$ in the versatile case), with the canonical hemodynamic response function at the onsets of the corresponding stimuli (these convolved regressors are often referred to as "parametric modulations" in the fMRI community).

Additionally, we included in the encoding model other regressors corresponding to factors of no-interest in this study, which we removed during testing to evaluate the specific encoding of probability or confidence. The regressors of no-interest included six motion regressors, and the following.

- Parametric modulations of stimulus onsets associated with factors other than probability and confidence:
  - A constant
  - The Shannon surprise induced by the stimulus, $-log\,p(s)$, where $s$ is the stimulus value and $p(s)$ is the normative probability estimate for that stimulus value given the previously observed stimuli

  - The Shannon entropy of the outcome implied the normative probability estimate, $-p*\log(p) - (1-p)*\log(1-p)$

- Parametric modulations of response screen onsets modeling reporting periods (including, for each period, a response screen for probability and another for confidence):
  - A constant
  - The normative estimate
  - The estimate reported by the participant

The inclusion of surprise and entropy as covariates was intended to improve specificity and prevent misidentifying codes for surprise or entropy as codes for probability or confidence (see Supplementary Fig. 5).

Finally, we applied the same temporal preprocessing to all regressors as we applied to the fMRI signal (detrending, filtering, z-scoring across sessions, and session-wise demeaning, see MRI data preprocessing section).

At the voxel level, the fMRI activity predicted by the model after preprocessing $\hat{y}_{fMRI}$ is written as $\hat{y}_{fMRI} = w\,\dot{x} + \boldsymbol{w_n}\,\boldsymbol{n}$ for the linear model, and $\hat{y}_{fMRI} = \boldsymbol{w}\,\dot{f}(x) + \boldsymbol{w_n}\,\boldsymbol{n}$ for the versatile model, where "·" represents the convolution and temporal preprocessing operations, $\boldsymbol{w}$ and $\boldsymbol{f}$ are the vectors $[w_1, ..., w_k]$ and $[f_1, ..., f_k]$, $\boldsymbol{n}$ is the vector comprising all preprocessed regressors of no-interest, and $\mathbf{w_n}$ is their associated weights vector (bold symbols denote vectors as opposed to scalar quantities).

The encoding models were implemented using Python, NumPy (https://numpy.org), and nilearn (https://nilearn.github.io).

## Evaluation of the encoding models

We evaluated the ability of the encoding models to predict fMRI data for a given participant using leave-one-session-out cross-validation: three sessions were used for fitting the model, and the fourth, left-out session, was used to test the fitted model. This procedure was repeated for each of the four possible choices of the left-out session, and the test scores obtained were averaged across the four left-out sessions.

**Fitting.** The weights of the encoding model were fitted using Ridge regression. The Ridge penalty was fixed for a given model in different voxels, and identical for different models having the same number of regressors. We optimized the Ridge penalty through simulations.

**Testing.** During testing, we calculated the fMRI activity predicted by the model using only the part of the model associated with the factors of interest (probability or confidence). (This is equivalent to replacing the regressors of no-interest with their mean value for the session, which is equal to 0 after session-wise demeaning.) As a score, we first calculated the $R^2$ obtained by comparing the model predictions with the actual fMRI data, using the sums-of-squares formulation[50]. We then calculated a null distribution of $R^2$ scores by injecting null predictions into the $R^2$ calculation. These null predictions were obtained by replacing the true stimulus sequence seen by the participant with another sequence randomly generated according to the task generative process, for 100 generated sequences. Finally, we calculated the score we call $z$-$R^2$ by standardizing the $R^2$ score obtained for the true sequence by the mean, $\mu_0(R^2)$, and standard deviation, $\sigma_0(R^2)$, of the null distribution: $z$-$R^2 = (R^2 - \mu_0(R^2))/\sigma_0(R^2)$.

Models were fitted and tested using Scikit-Learn (https://scikit-learn.org).

## Simulation of encoding models

We conducted simulations to verify that our procedure, applied in our experimental protocol, was able to detect and differentiate a neural encoding of probabilities or confidence, monotonic or non-monotonic. For this purpose, experimental data were generated assuming a certain encoding model, and the generated experimental data were analyzed with other encoding models. As for participants, each generated experiment consisted of four sessions, with one sequence of stimuli per session, generated according to the task process. The distributions of normative probability and confidence reports used in the simulation match, by construction, the ones used in the analysis of the fMRI data. Noisy fMRI activities were then generated for each session and a certain number of simulated voxels assuming a certain generative model of neural activity, which corresponded to one of the four encoding models presented above (encoding of probability or confidence estimates, according to a linear or a versatile encoding model).

The procedure for generating fMRI activity was as follows. For each simulated voxel, we randomly generated weights for the generative model by drawing each weight uniformly in [−0.5, 0.5]. We generated the "signal" component of the fMRI activity in accordance with the generative model, by calculating the weighted sum of the corresponding fMRI regressors as described in the above section on encoding models. We then injected Gaussian white noise with power equal to nine times that of the signal, in order to obtain a signal-to-noise ratio of 10% signal to 90% noise. This produced a set of fMRI activities for each generated experiment and each possible generative model.

For each generated experimental data set, the procedure presented in the above section was used to fit the encoding models and evaluate their ability to predict the simulated fMRI data (as

subsequently done for the participants' fMRI data). This produced one $z$-$R^2$ score per voxel, fitted model, generative model, and generated experiment. We averaged the scores obtained across one hundred experiments and one hundred simulated voxels to obtain an average $z$-$R^2$ score for each possible generative model × fitted model pair, resulting in a 4 × 4 matrix, shown in Fig. 3B.

This analysis was also performed by splitting the versatile model into one with Gaussian basis functions and one with sigmoid basis functions to produce the matrix shown in Supplementary Fig. 3.

We also carried out this analysis by adding to the encoding models a linear model of entropy to produce the matrix shown in Supplementary Fig. 5A, B. The same procedure as for the other models was used to generate or fit the fMRI data, except that in the case of the linear model of entropy, entropy was not included as a regressor of no interest, since entropy is a regressor of interest.

The simulations were implemented using Python and Numpy.

## MRI data acquisition
**Equipment.** The MRI scanner was a Siemens MAGNETOM 7 Tesla at the NeuroSpin center (CEA Saclay, France), with whole-body gradient and 32-channel head coil by Nova Medical.

At the beginning of each participant's data acquisitions, a $B_0$ map was acquired and loaded in the console to perform shimming first of the whole brain, then in an interactive fashion on the occipito-parietal cortex. A $B_1$ map was then acquired and the intraparietal sulcus values were used to compute a voltage to be used as the system's reference voltage.

**Functional MRI acquisition.** Whole-brain T2*-weighted fat-saturation functional volumes with 1.5 mm isotropic voxels were acquired using a multi-band accelerated echo-planar imaging (EPI) sequence (Moeller et al. 2010; https://www.cmrr.umn.edu/multiband/). The sequence parameters were: multi-band factor = 2; GRAPPA acceleration factor = 2 (IPAT); partial Fourier = 7/8 (PF); matrix = 130 × 130, number of slices = 68, slice thickness = 1.5 mm, repetition time = 2 s (TR); echo time = 22 ms (TE); echo spacing = 0.71 ms (ES); flip angle = 68° (FA); bandwidth = 1832 Hz/px (BW); phase-encoding direction: anterior to posterior. The reference volume for the GRAPPA reconstruction collected at the beginning of each sequence used the Gradient Recalled Echo (GRE) option provided by the multiband EPI sequence.

Before each session, two single-band functional volumes were acquired with the above parameters except that they had opposite phase-encoding directions. This was later used for distortion correction (see below in *Preprocessing*).

**Anatomical MRI acquisition.** Whole-brain T1-weighted anatomical images with 0.75 mm isotropic resolution were acquired using an MP2RAGE sequence. The sequence parameters were: GRAPPA acceleration factor = 3 (IPAT), partial Fourier = 6/8 (PF), matrix = 281 × 300, repetition time = 6 s (TR), echo time = 2.96 ms (TE), first inversion time = 800 ms ($TI_1$), second inversion time = 2700 ms ($TI_2$), echo spacing = 6.9 ms, flip angle 1 = 4° ($FA_1$), flip angle 2 = 5° ($FA_2$), bandwidth = 240 Hz/px (BW).

## MRI data preprocessing
The functional volume slices were corrected for slice-timing with respect to the slice acquired in the temporal middle of a volume acquisition. The functional volumes were corrected for motion using rigid transformations, and co-registered to the anatomical image. Session-wise distortion correction was applied to the functional volumes using FSL apply_topup, after having estimated a set of field coefficients for the session using FSL TOPUP with the two-single band volumes with opposite phase-encoding directions acquired before that session.

Prior to encoding and decoding analyses, the fMRI time series of each voxel and session were temporally detrended and high-pass filtered at a cutoff frequency of 1/128 Hz. The series from the four sessions were then concatenated in order to z-score the fMRI data across the four sessions for each voxel (this was done for numerical convenience). Note that we did not z-score the fMRI data per session because under the versatile encoding model, it is expected that the variance of the fMRI signal should change from session to session depending on the probabilities and confidence levels estimated during each session.

Note that the preprocessing pipeline does not contain a spatial smoothing step before the estimation of the encoding model, unlike what is customary in fMRI analysis, in order to preserve the spatial resolution of the data as much as possible. Spatial smoothing would mix the signal from neighboring voxels, at the risk of masking an effect of probability. For instance if neighboring voxels have monotonic tuning curves with opposite slopes, or non-monotonic tuning curves with peaks at different probability bins, then the average tuning curve would become flatter, reducing the fraction of variance in the (smoothed) fMRI signal that is explained by probability.

Slice-timing correction, motion correction, and co-registration were done using tools from SPM12. Distortion correction was done using tools from FSL. SPM12 and FSL were called from Python using the NiPype module (https://nipype.readthedocs.io). Further preprocessing was done using Python and the NumPy, SciPy (https://scipy.org) and nilearn (https://nilearn.github.io) packages.

## Conversion of volumetric data into cortical surface data
Here we detail the processing steps we used throughout this study to project participant-level volumetric data, such as the $R^2$ and $z$-$R^2$ scores computed from the fMRI data (see section on encoding model evaluation), onto the cortical surface, and to bring them into a common space across participants. These processing steps were performed using FreeSurfer (https://surfer.nmr.mgh.harvard.edu) and the Python interface to FreeSurfer commands provided by the NiPype module. The resulting surface data were then analyzed by working directly with the numerical arrays in Python, unless otherwise stated.

For each participant, the cortical surface was reconstructed from the acquired high-resolution anatomical MRI image by running the FreeSurfer command "recon-all". These reconstruction data were then used to project other volumetric data calculated from the functional data onto the cortical surface, working in the participant's native space. This projection was performed using the "mri_vol2surf" command. The surface data were then normalized, i.e., resampled to be brought into a common space: the "fsaverage" template of FreeSurfer, version 7 high-resolution (163,842 vertices per hemisphere), and were spatially smoothed with a Gaussian kernel of 3 mm full width at half maximum. The resulting data are surface maps containing one data point per vertex (a vertex is the surface equivalent of a voxel in the volume, called vertex because vertices are assembled to define polygons that together form a three-dimensional mesh of the cortical surface).

## Group-level statistical maps
The participants' $z$-$R^2$ surface maps were grouped together and a one-sample t-test against zero was performed at the group level. The resulting $p$-value maps were thresholded at the vertex level with a threshold of $p < 0.001$, and corrected for multiple comparisons with a family-wise error rate of $p_{FWE} < 0.05$ using FreeSurfer's Monte Carlo simulation-based cluster correction with a vertex-wise cluster-forming threshold of $p < 0.001$.

## Definition of vertices of interest for characterization
To ensure the reliability of the tuning curves used for characterization, we defined a set of vertices of interest for which the encoding signal

was sufficiently strong. One set was defined per type of estimate and per participant. Additionally, the number of vertices was adjusted depending on the availability of vertices with a strong enough signal.

The definition was done in two steps: first at the group level, and then at the participant level. At the group level, a region of interest was defined as the union of parcels from the HCP-MMP1.0 atlas[34] containing the significant clusters found in the group-level statistical maps obtained for the corresponding estimate type (clusters shown on Figs. 4 and 5 and listed in Table 1 and Supplementary Table 1). The vertices of interest for each participant were then defined within the group-level region, using the following steps. The $R^2$ scores obtained with the encoding model were converted into $p$-values in each participant and vertex. For each vertex and participant, a $p$ value was calculated from the $R^2$ score as the probability of obtaining a value at least as large in a distribution of $R^2$ values obtained when the fMRI data is replaced with white noise (empirically calculated for each participant with 1,000,000 noise samples). Finally, the vertices with FDR-corrected $p < 0.05$ were selected as vertices of interest (controlling the False Discovery Rate using the Benjamini–Hochberg procedure).

As mentioned in the results section, we verified the reliability of our estimations within the defined vertices of interest using a test-retest procedure, by fitting the weights of the versatile encoding model on two independent halves of the data and comparing the estimated weights between the two halves.

## Characterization of tuning curves

Three characteristic measures were defined and applied to the tuning curves estimated for each vertex of interest.

*1) Proportion of tuning curves maximized at non-extreme values.* For each tuning curve, we took the input value at which the curve was maximized (i.e., the argmax of the tuning curve function) and treated it as non-extreme if it fell between the lower and upper bounds of the domain with a margin of at least twenty percent relative to the span of the domain. The proportion refers to the proportion of vertices out of all vertices of interest within the participant.

*2) Non-monotonicity index.* Mathematically speaking, a differentiable function is said to be monotonic if its derivative remains of the same sign over its domain. By extension, we defined the monotonicity index $m(f)$ of a function as the absolute value of its average derivative, normalized such that the index of any purely monotonic function is equal to 1. The non-monotonicity index $n(f)$ was $1 - m(f)$. It is calculated by the formula $n(f) = 1 - |f(x_{max}) - f(x_{min})| / (f_{max} - f_{min})$, where $x_{min}$ and $x_{max}$ are the lower and upper bounds of the domain, and $f_{min}$ and $f_{max}$ are the minimum and maximum of the function over the domain, respectively. The indices calculated for each tuning curve function were averaged across the vertices of interest to obtain a single value per participant. Note that $m(f) = 1$ is a necessary condition for a monotonic function, but not a sufficient one (non-monotonic functions can yield $m(f) = 1$).

*3) Nonlinearity index.* As for non-monotonicity, we defined for nonlinearity a continuous index $v(f)$ which measures the overall deviation of a tuning curve function from linearity. This was calculated by fitting a linear function to the tuning curve via linear regression and calculating the proportion of variance unexplained by the linear fit (i.e., residual): $v(f) = SS_{lin,res} / SS_{tot} = 1 - R^2_{lin}$, where $SS_{lin,res}$, $SS_{tot}$, and $R^2_{lin}$ are the residual sum of squares, total sum of squares, and $R^2$ of the linear regression, respectively. To compute group-level statistics, as for the non-monotonicity index, the nonlinearity indices for each tuning curve were first averaged across vertices of interest at the participant level. The group analysis was then conducted on the average indices of each participant. In the discussion, we reported the proportion of tuning curves exhibiting more than one peak. We calculated this proportion at the participant level by first computing the number of peaks that each tuning curve exhibited and then computing the proportion of tuning curves for which this number was strictly more than one. We

computed the number of peaks of each tuning curve using the peak detection algorithm from scipy "scipy.signal.find_peaks". We used fairly conservative conditions for a local maximum to be considered a peak, imposing, after normalizing the curve between 0 and 1, a minimum peak height of 0.7 and a minimum prominence of 0.125 (the height measures the peak's vertical distance from 0 while the prominence measures its vertical distance from the surrounding baseline).

## Decoding of probability and confidence

The same method was used for decoding probability and confidence; here, we explain the method in the case of probability. Decoding was performed at the level of each session and participant, and it leveraged the methods used for the encoding analysis. The encoding model characterized a pattern of voxel responses to (overlapping) bins of probability as a set of regression weights assigned to Gaussian basis functions. The decoder aimed to identify the probability bin corresponding to the response pattern measured across several voxels in a test session. Decoding accuracy was assessed with leave-one-session-out cross-validation. The encoding model was fitted (with 5 basis functions instead of 10) separately in the training set (three sessions) and the left-out session. The decoder assigned a probability bin to a given response pattern on the test set by identifying the probability bin eliciting the most similar response pattern in the training set. More precisely, the decoder used the correlation distance $D(j, k)$ between the response patterns corresponding to the $j$-th probability bin in the test session and the $k$-th bin in the training set, and looked for the $k$ that minimized $D$ for a given $j$. The correlation distance is one minus the Pearson correlation[51] qualitatively similar results, although inferior, were obtained with the Euclidean distance.

The decoder was applied to selected voxels in each parcel of the Glasser et al. atlas[34]. This atlas is available in the fsaverage template used for anatomical normalization (https://figshare.com/articles/HCP-MMP1_0_projected_on_fsaverage/3498446). The atlas was projected onto the native anatomical surface of each participant using the inverse normalization transform, and projected back into their native volume, both with FreeSurfer. The voxels corresponding to each parcel in the functional images were identified based on the coregistration of functional and anatomical images. Decoding was applied to the 100 voxels with the largest $z$-$R^2$ value, estimated within the training sessions only.

For each participant, decoding accuracy was assessed for each probability bin as the fraction of correctly assigned probability bins (chance level is 1 out of 5 possibilities) on each left-out session, and averaged across the left-out sessions. Some probability bins concerned fewer than 5% of stimuli in some test sessions, which we deemed unreliable; the corresponding session-level accuracy was omitted from the average across sessions. More precisely, we considered that a stimulus fell in a given bin if its probability elicited more than 10% of the maximum value of the corresponding basis function. The significance of decoding accuracy was assessed, for each parcel, with a two-sided t-test against chance-level accuracy at the group level, and corrected for multiple comparisons across parcels with false discovery rate (0.05) correction (Benjamini-Hochberg procedure).

## Analysis of representational dissimilarity matrices (RDMs)

The same method was used for probability and confidence; here, we explain the method in the case of probability. We analyzed the distance matrices $D(j, k)$ used for the decoding process, which are also known as RDM. For each participant the RDM of each parcel was the average RDM obtained across cross-validation folds (ignoring the rows $j$ of each RDM corresponding to probability bins concerning fewer than 5% of stimuli in a session). Note that since RDMs were estimated with leave-one-session-out cross-validation, they are not symmetric and their diagonal is not 0. The empirical RDMs were compared to the RDMs of different neural codes, namely, the identity RDM that should

arise from a highly non-monotonic code, and the graded RDM that should arise from a linear (and more generally monotonic) code. We estimated the regression weights corresponding to each theoretical RDM in a multiple regression analysis; theoretical RDMs were z-scored to make the regression weights commensurable.

## Reporting summary

Further information on research design is available in the Nature Portfolio Reporting Summary linked to this article.

## Data availability

The behavioral data and raw preprocessed MRI data used in this study are available at https://gin.g-node.org/zamor/EncodeProb_7T.

## Code availability

All analysis code to reproduce the reported results and figures from the shared data, as well as a notebook that explains the methods, are available at https://github.com/TheComputationalBrain/fmri_code_event_probability. The following Python packages were used: nilearn 0.11.0, numpy 1.26.4, scikit-Learn 1.5.1. Archived code as at time of publication: https://doi.org/10.5281/zenodo.17152441[52].

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

## Acknowledgements
This work was supported by funding from the French National Research Agency (ANR grant #18-CE37-0010-01) and from the European Research Council (ERC grant #947105) to F.M. C.F. was supported by a PhD fellowship from ENS Paris-Saclay (France). We thank Zaineb Amor for technical assistance with data sharing, and Alice Hodapp and Aaron Greenhouse-Tucknott for comments on the manuscript.

## Author contributions
C.F.: conceptualization, methodology, software, validation, formal analysis, writing—original draft, writing—review and editing, visualization; TB: conceptualization, methodology, software, investigation; S.D.: methodology, software; BT: conceptualization, methodology, writing—review and editing; E.E.: conceptualization, methodology, writing—review and editing; F.M.: conceptualization, methodology, software, formal analysis, resources, writing—original draft, writing—review and editing, visualization, supervision, project administration, funding acquisition.

## Competing interests
The authors declare no competing interests.
