## [Transparent Peer Review File · Nature Communications]

A non-monotonic code for event probability in the human brain

Corresponding Author: Dr Florent Meyniel

Version 0:

Reviewer comments:

Reviewer #1

(Remarks to the Author)

The authors have been responsive to my comments and have addressed all the issues I raised with new analyses, plots, and/or modifications of the text. These changes have strengthened the manuscript and it now provides a nuanced exposition and discussion of the results. Only one minor issue remains that the authors should address.

In response to my original comment 5, the authors state that they weakened the claims that the code for confidence is linear. They did so by adding one sentence in the introduction to define that by “linear” they only mean “smooth monotonic”. This qualification is necessary given the low correlation ($r=0.16$) of the normative confidence “predictions” they use for the fMRI analysis and the actual confidence reports, and the fact that in about one third of their participants, a non-linear model would actually provide a better fit (see their new AIC analyses). However, the ensuing manuscript in many places still very much emphasizes that the code for confidence is linear (for example, they claim “In contrast, the tuning curves for confidence were essentially linear.”, but there are several other places with similar statements). This will lead to misunderstandings in future readers, despite the one qualifying sentence in the introduction. The authors should follow my initial recommendation and be explicit in the discussion that their data cannot provide evidence for “linearity” of the confidence code, only for “smooth monotonicity”.

(Remarks on code availability)

Reviewer #2

(Remarks to the Author)

As previously, I like the idea behind the study and am impressed with the technical strength of the work.

I appreciate the authors' work to address my points.

WRT my points raised in the initial review –

Conceptual – I appreciate that the authors have clarified the ms on the issue of ‘pure probability’, making clear that “probabilities can be about many different things”. However, I still think there are some over-strong claims in the manuscript. The abstract states that “While previous studies have shown that neural activity in several brain regions correlates with probability-related factors such as surprise and uncertainty, similar correlations have not been found for probability,” apparently dismissing multiple huge bodies of work on this topic. For example Mike Shadlen’s work over 25 years in which the code for probability is studied (probability being operationalized as evidence for a decision) or Wolfram Schultz’s work in which the dopamine signal at the CS tracks the probability of reward. I don’t agree that these operationalizations of probability are less valid than the authors’ definition of probability as the generative frequency of L vs R tilted gratings.

Rather than claiming to be the first to find a code for probability in the brain, would a more reasonable framing be the following: the probability of stimuli or outcomes could be encoded by a variety of neural codes including, linear, monotonic nonlinear (such a log-) codes and 'basis set' type codes such as cosine tuning curves. Traditional fMRI methods are not sensitive to all of these neural-level codes, because they assume voxel responses are linear and this will not be the case if a voxel contains a mixture of monotonic nonlinear or basis set tuned neurons. The manuscript proposes a new method that is equally sensitive to all coding types and is able to find encoding of both expected stimulus orientation, and confidence in multiple brain regions. The recovered voxel tuning curves are consistent with codes at the neural level of the 'basis set' type.

Minor conceptual -

The counterbalancing of the order of presentation of the probability and confidence probes seems quite important to me – I suggest to mention it when the task is initially described (top of p3), whereas now I think it is only mentioned in the detailed Methods section.

Technical –

On a general point, this manuscript is quite technical and perhaps lacking some explanation of *why* these particular approaches were needed or informative. As an expert reviewer I have sometimes struggled to work out exactly what is meant where technical terms are used, or what the implications are of the analysis methods chosen. I have not always been clear what the authors are really inferring about the brain on the basis of these complex analyses (beyond 'this one fits the data better' – why does it matter? What do we infer about the brain?). I admit that points may pass me by which would be clear to someone working very directly with these methods but nonetheless I must be closer in field and expertise to the authors than 99% of the journal's readership, so it is probably fair to say that the potential impact of the paper would be improved by spelling out these points most clearly for a more general audience. Maybe the authors could get a non-mathematically trained colleague to read through and ask questions, then use this to work out what explanatory text may be needed.

My previous point:

- Is the linear model a reasonable control? This is predicting (univariate) voxel activity as a linear function of the probability that the grating is oriented left (rather than right). But the brain could be full of voxels with a linear code for the probability of left OR right, and if these populations were intermixed, you would see no linear code for probability overall.

Firstly, I'm sorry for the confusion – I should have said that the brain could be full of intermixed *neurons* with linear codes, not voxels.

Additionally, I think there is some confusion on my part in terms of what is meant here by a linear code. One interpretation would be of a linear code as a simple straight line relationship between the encoded quantity (probability/confidence) and activity in the voxel or neuron.

$$Y = wx + \text{error} \quad (1)$$

Another more general interpretation would be a linear code as linear modelling,

$$Y = w \cdot f(x) + \text{error} \quad (2)$$

Where f is a single function, for example $Y = w \cdot \cos(x)$ or $Y = w \cdot \log(x)$

... noting that these are both distinct from the basis set model in which

$$Y = \sum(w_i \cdot f_i(x)) + \text{error} \quad (3)$$

I see that form (1) is the definition of linear code used and agree that if this strict form is used, you cannot get a non-linear voxel response from linear cell responses.

However, I stand by my point that non-linear monotonic codes such as $Y = w \cdot \log(x)$ are conceptually quite similar to the straight-line relationship in (1), which I think the authors agree with.

But if that is the case, we still have the issue that the non-linear voxel code could just reflect a mixture of cells with those non-linear monotonic codes (eg cells that respond as $Y = w \cdot \log(p_{\text{Right}})$ and $Y = w \cdot \log(p_{\text{Left}})$). This exact example would be a result in which a non-linear and non-monotonic voxel code is found, but the code at the neural level is exactly the same as those reported many times previously (eg in Shadlen's work).

I take the point that the analysis is only of tuning curves at the voxel level not the cellular level. But surely the important inference is about the code at the cellular level not the voxel level, since cells are units of the brain and voxels are only units of your image of the brain...? And if no inference is being made about tuning at the cellular level, should the introduction then avoid discussing studies such as Yang and Shadlen which are about tuning at the cellular level, because it is acknowledged that no conclusions can be drawn about the neuronal level?

(Remarks on code availability)

Reviewer #3

(Remarks to the Author)

The authors have addressed a number of my previous concerns. I believe the resulting changes and clarifications have substantially improved the manuscript. I still found that some of changes, in particular with exposition, to be incomplete.

- While the authors have revised to manuscript to acknowledge the presence of alternative explanations, they still maintain the assumption that the responses captured are related to probability without qualification in much of the introduction, results, and even title. I think these needs to be revised to be consistent with the conclusion that, while the authors provide evidence consistent with probability encoding, there is no definitive evidence for a “neural code” per se, rather responses that are consistent with those generated via a neural code, but also other possible underlying mechanisms.

- The inclusion of the entropy model in comparison to confidence and probability Fig. S4 was helpful, and provides evidence that confidence and probability responses cannot be reduced to entropy. At the same time, related to the previous point, I don't think this analysis is sufficient to rule out entropy as an alternative explanation, given the similarity in response patterns and the lack of experimental manipulation that can dissociate the two models. Similar to above, I don't think this is fatal to the paper but the limitations need to be clearly acknowledged.

- I'm not sure I understood the authors' response to my question about the range of confidence responses. According to Fig. 1B, participants' confidence responses ranged from ~0.7 to ~0.8, with the scale ranging from 0-1. If that is the case, participants' confidence reports are quite compressed, regardless of their correlation with the normative model.

(Remarks on code availability)

The code was quite well organized. I did not run it but it is one of the better code repos I've seen.

Version 1:

Reviewer comments:

Reviewer #1

(Remarks to the Author)

The authors have again been very responsive to my comment (and those of the other reviewers) and have changed the introduction and discussion of the results to more precisely characterize the aim of the study and the conclusions that can be drawn from the results. They have also added a code notebook and discuss the utility of their method for studying probability (as opposed to just the implications of their specific findings), which is really useful for the field. I think this version of the manuscript is vastly improved and should be published. It will find many interested readers.

(Remarks on code availability)

I have not had time for a very thorough review of the code, but there are clear instructions on how to run it, the code itself is well structured, and it is very useful both for reproducing the figures of the paper and for using this analysis method for other purposes. This is what authors should provide!

Reviewer #2

(Remarks to the Author)

All my comments have been addressed

I would like to thank the authors for their work in responding to my comments

(Remarks on code availability)

Reviewer #3

(Remarks to the Author)

I thank the authors for their responses, but their response to the last point I raised regarding confidence was rather confusing. First, the authors refer to Supplementary Fig 6 for a histogram of subjective confidence, but that only shows the simulated RSA matrices. I believe the authors are instead referring to Supplementary Fig 1. However, in that figure, the skew in the subjective confidence measures are very noticeable, and only “spans the full range” in the technical sense. The same issue is present for normative confidence. Given this, I don't see how the authors' response address the issue of range compression for the confidence measures.

(Remarks on code availability)

No concerns.

Point-by-Point Response to Reviewers

Here is a summary of the main changes:

- We improved the clarity and impact for a broader readership (prompted by Reviewer #2), with several additions:
 - A new main figure (Fig 2) that explains the basics of our methods, in particular the use of a function basis set to model fMRI data generated with an unknown tuning curve
 - A notebook, both as a PDF and an interactive python version, that explains our methods (corresponding to Fig 2, Fig 3A and Fig 6A). When used in lab meeting to explain the paper, this notebook greatly enhanced comprehension. This notebook should make our methods easy to understand and reuse.
 - Several edits in the Results section, following a close read by two test colleagues external to this project and without an extensive background in fMRI data analysis.
 - We have changed the Introduction (see new second paragraph) to make our goal clear more up front in the paper, and avoid confusion regarding other types of probabilities not studied in our article (Reviewer #2)
- To increase the impact of the paper to a broader audience, in particular to neuroscientists who do not use neuroimaging, we have changed a bit our terminology: when presenting our results, we no longer talk about “neural code” more more simply of “code”, as the former has often been associated with the single-neuron level in the neuroscience literature, rather than the millimeter scale probed by fMRI (Reviewer #2 and #3)
- Again in terms of terminology, we have replaced “linear code” with “monotonic code” throughout the paper. This is more mathematically correct, and it better captures our goal (Reviewer #1)
- We have cleaned up our code (although Reviewer #3 thought it was already better than usual), and shared it publicly.
- We have uploaded our behavioral and MRI data on a public repository (and provide the link in the paper)

Below is a point-by-point response to the reviewers.

Reviewer #1

The authors have been responsive to my comments and have addressed all the issues I raised with new analyses, plots, and/or modifications of the text. These changes have strengthened the manuscript and it now provides a nuanced exposition and discussion of the results. Only one minor issue remains that the authors should address.

In response to my original comment 5, the authors state that they weakened the claims that the code for confidence is linear. They did so by adding one sentence in the introduction to define that by “linear” they only mean “smooth monotonic”. This qualification is necessary given the low correlation ($r=0.16$) of the normative confidence “predictions” they use for the fMRI analysis and the actual confidence reports, and the fact that in about one third of their participants, a non-linear model would actually provide a better fit (see their new AIC analyses). However, the ensuing manuscript in many places still very much emphasizes that the code for confidence is linear (for example, they claim “In contrast, the tuning curves for confidence were essentially linear.”, but there are several other places with similar statements). This will lead to misunderstandings in future readers, despite the one qualifying sentence in the introduction. The authors should follow my initial recommendation and be explicit in the discussion that their data cannot provide evidence for “linearity” of the confidence code, only for “smooth monotonicity”.

Our analysis relates the normative probability and confidence to neural activity, so the possible non linearity of subjective confidence with respect to normative confidence does not alter our conclusion regarding the neural code for normative probability and confidence.

This being said, we realized that it is confusing to use the phrase “linear code” where in fact we are merely interested in a smooth monotonic code, even after including a few clarification sentences. We thus opted for a more radical option, and **we have changed “linear code” into “monotonic code”, and “nonlinear code” with “non-monotonic code” throughout the paper.**

Reviewer #2:

As previously, I like the idea behind the study and am impressed with the technical strength of the work.

I appreciate the authors’ work to address my points.

WRT my points raised in the initial review –

Conceptual – I appreciate that the authors have clarified the ms on the issue of ‘pure probability’, making clear that “probabilities can be about many different things”. However, I still think there are some over-strong claims in the manuscript. The abstract states that “While previous studies have shown that neural activity in several brain regions correlates with probability-related factors such as surprise and uncertainty, similar correlations have not been found for probability,” apparently dismissing multiple huge bodies of work on this topic. For example Mike Shadlen’s work over 25 years in which the code for probability is studied (probability being operationalized as evidence for a decision) or Wolfram Schultz’s work in which the dopamine signal at the CS tracks the probability of reward. I don’t agree that these operationalizations of probability are less valid than the authors’ definition of probability as the generative frequency of L vs R tilted gratings.

Rather than claiming to be the first to find a code for probability in the brain, would a more reasonable framing be the following: the probability of stimuli or outcomes could be

encoded by a variety of neural codes including, linear, monotonic nonlinear (such a log-) codes and 'basis set' type codes such as cosine tuning curves. Traditional fMRI methods are not sensitive to all of these neural-level codes, because they assume voxel responses are linear and this will not be the case if a voxel contains a mixture of monotonic nonlinear or basis set tuned neurons. The manuscript proposes a new method that is equally sensitive to all coding types and is able to find encoding of both expected stimulus orientation, and confidence in multiple brain regions. The recovered voxel tuning curves are consistent with codes at the neural level of the 'basis set' type.

The reviewer is quite right that the current version of the introduction does not do justice to the large literature on reward probability, and that when this literature is ignored, the claim of novelty is not legitimate. In fact, our claim is about the probability of affectively neutral event (which excludes reward probability or the probability of a decision being correct), but this clarification appeared only in the penultimate paragraph of the introduction. **The revised version makes this point clear upfront with a new dedicated paragraph that is the second of the introduction.**

Minor conceptual -

The counterbalancing of the order of presentation of the probability and confidence probes seems quite important to me – I suggest to mention it when the task is initially described (top of p3), whereas now I think it is only mentioned in the detailed Methods section.

We now mention this feature of the task design in the second paragraph of the Results section.

Technical –

On a general point, this manuscript is quite technical and perhaps lacking some explanation of *why* these particular approaches were needed or informative. As an expert reviewer I have sometimes struggled to work out exactly what is meant where technical terms are used, or what the implications are of the analysis methods chosen. I have not always been clear what the authors are really inferring about the brain on the basis of these complex analyses (beyond 'this one fits the data better' – why does it matter? What do we infer about the brain?). I admit that points may pass me by which would be clear to someone working very directly with these methods but nonetheless I must be closer in field and expertise to the authors than 99% of the journal's readership, so it is probably fair to say that the potential impact of the paper would be improved by spelling out these points most clearly for a more general audience. Maybe the authors could get a non-mathematically trained colleague to read through and ask questions, then use this to work out what explanatory text may be needed.

We have asked two colleagues, less versed in the methods used in the paper, to carefully read the paper. Based on their comments, we have rephrased some sections of the paper to enhance clarity and impact for a more general audience. **One key change is the addition of a new figure (now Fig. 2) and an interactive Python notebook (Supplementary material) that explain the methods**, in particular, the use of a basis set to model a (fMRI) time series

in response to a quantity of interest (here, probability or confidence), and to estimate the corresponding tuning curve.

My previous point:

- Is the linear model a reasonable control? This is predicting (univariate) voxel activity as a linear function of the probability that the grating is oriented left (rather than right). But the brain could be full of voxels with a linear code for the probability of left OR right, and if these populations were intermixed, you would see no linear code for probability overall.

Firstly, I'm sorry for the confusion – I should have said that the brain could be full of intermixed *neurons* with linear codes, not voxels.

Additionally, I think there is some confusion on my part in terms of what is meant here by a linear code. One interpretation would be of a linear code as a simple straight line relationship between the encoded quantity (probability/confidence) and activity in the voxel or neuron.

$$Y = wx + \text{error} \quad (1)$$

Another more general interpretation would be a linear code as linear modelling,

$$Y = w.f(x) + \text{error} \quad (2)$$

Where f is a single function, for example $Y=w.\cos(x)$ or $Y=w.\log(x)$

... noting that these are both distinct from the basis set model in which

$$Y = \sum(w_i.f_i(x)) + \text{error} \quad (3)$$

I see that form (1) is the definition of linear code used and agree that if this strict form is used, you cannot get a non-linear voxel response from linear cell responses.

However, I stand by my point that non-linear monotonic codes such as $Y=w.\log(x)$ are conceptually quite similar to the straight-line relationship in (1), which I think the authors agree with.

Model (2) pointed out by the reviewer does not correspond in general to a monotonic code (and even less a linear code), because the fMRI activity Y is not a monotonic function of the quantity of interest x in general (unless f is monotonic).

The key point of the paper is that the model (3), corresponding to the versatile encoding model in the paper, can be used to model a signal generated by model (2) even when the function f is unknown.

We have implemented several changes in response to these points.

- We have replaced “linear code” with “monotonic code”. We very much agree that a form such as $\log(x)$ is conceptually similar to simply a straight line. In fact, we are

not interested in strict linearity, a point that was already mentioned in the paper, and that is now completely clear after changing terminology.

- We have added a new Figure 2, and a Python notebook as supplementary material, and improved Fig 3A (previously Fig 2A) to better explain the methods. The example used in (the new) Fig 2 is actually the one pointed out by the reviewer, in which the tuning curve f is a cosine (plus a linear trend), and thus, the code is non-monotonic. The versatile encoding model accounts for this generative model very well, as shown in the figure.

But if that is the case, we still have the issue that the non-linear voxel code could just reflect a mixture of cells with those non-linear monotonic codes (eg cells that respond as $Y=w.\log(pRight)$ and $Y=w.\log(pLeft)$). This exact example would be a result in which a non-linear and non-monotonic voxel code is found, but the code at the neural level is exactly the same as those reported many times previously (eg in Shadlen's work).

The simulation below illustrates the point of the reviewer. It actually corresponds to a special case of the versatile encoding model with only two basis functions:

$$f_1(x)=\log(x)$$

$$f_2(x)=\log(1-x)$$

This generative model can produce both monotonic and non monotonic tuning curves at the voxel level. This specific model could be tested against other models; a quick inspection of the tuning curves observed in our dataset suggests that this specific model would not be ideal; for instance it cannot predict the existence of tuning curves with more than one peak at non extreme probabilities. However, this model could be enriched, including more basis functions corresponding to various distortions of the log function and become a radial basis set with equivalent approximation properties as the Gaussian and sigmoidal basis set tested in the paper.

I take the point that the analysis is only of tuning curves at the voxel level not the cellular level. But surely the important inference is about the code at the cellular level not the voxel level, since cells are units of the brain and voxels are only units of your image of the

brain...? And if no inference is being made about tuning at the cellular level, should the introduction then avoid discussing studies such as Yang and Shadlen which are about tuning at the cellular level, because it is acknowledged that no conclusions can be drawn about the neuronal level?

Our focus is the voxel level indeed. We have added a note in the Introduction pointing to the Discussion regarding implications of the results at the single-neuron level. We prefer to keep references that pertain to the single-neuron level because it would be unfortunate to ignore neuroscientific work done at other scales.

Importantly, the results at the voxel level have implications about the single-neuron level, even though there is no one-to-one mapping between these levels. **We have expanded the paragraph of the discussion dedicated to the implications of the results at the neural level.**

Reviewer #3:

The authors have addressed a number of my previous concerns. I believe the resulting changes and clarifications have substantially improved the manuscript. I still found that some of changes, in particular with exposition, to be incomplete.

- While the authors have revised to manuscript to acknowledge the presence of alternative explanations, they still maintain the assumption that the responses captured are related to probability without qualification in much of the introduction, results, and even title. I think these needs to be revised to be consistent with the conclusion that, while the authors provide evidence consistent with probability encoding, there is no definitive evidence for a “neural code” per se, rather responses that are consistent with those generated via a neural code, but also other possible underlying mechanisms.

We have revised several sections of the papers, notably the title, introduction, results and discussion. We believe that the neural code can be studied at multiple scales, including the millimeter scale probed by fMRI. We nevertheless agree that the previous version may not have adequately discussed the implications of the results at the single-neuron level. We include a warning specifically on this in the introduction, and we have greatly expanded **a paragraph of the discussion dedicated to the implications of the results in the terms of single-neuron neural code** (see also responses to Reviewer #2). We also opted, in the revision, to **now simply write “code” rather than “neural code” when we present our results** (as the phrase 'neural code' has often been associated with the single-neuron level in the neuroscience literature).

- The inclusion of the entropy model in comparison to confidence and probability Fig. S4 was helpful, and **provides evidence that confidence and probability responses cannot be reduced to entropy**. At the same time, related to the previous point, I don't think this analysis is sufficient to rule out entropy as an alternative explanation, given the similarity in response patterns and the lack of experimental manipulation that can dissociate the two models. Similar to above, I don't think this is fatal to the paper but the limitations need to be clearly acknowledged.

This point is the first limitation that we discuss in the Discussion, in a dedicated and long paragraph. To acknowledge this point even more, we inserted a new sentence (“So it

seems that some of the neural correlates of probability we identified may actually be neural correlates of entropy”) and reused the reviewer’s phrase highlighted in bold.

- I’m not sure I understood the authors’ response to my question about the range of confidence responses. According to Fig. 1B, participants’ confidence responses ranged from ~0.7 to ~0.8, with the scale ranging from 0-1. If that is the case, participants’ confidence reports are quite compressed, regardless of their correlation with the normative model.

Several points are worth mentioning to address this concern.

- 1) Fig 1B reports subjective confidence averaged in bins of normative confidence. The correlation between the two being noisy (on average $r \sim 0.16$), this analysis exhibits a strong effect of regression toward the mean so that **Fig 1B does not provide the reader with a fair indication of the data range. The Supplementary Fig 6 shows the histogram of subjective confidence, which is more appropriate to assess the range.** This histogram spans the full range. Interestingly, it exhibits some concentration in the upper range, just as normative confidence (compare Supplementary Fig 1 panel A and B – see legend regarding the different units on the abscissa).
- 2) The neural data are not analyzed in terms of subjective confidence, but instead, in terms of normative confidence. So it is actually the distribution of normative confidence that should be considered when asking whether the confidence (or probability) range used in the task may favor a monotonic or non-monotonic code (which was mentioned in the original comment of the reviewer).
- 3) The simulation presented in Fig 3B, which uses the same distributions as the one used for the analysis of the actual fMRI data, shows that the identified model for confidence and probability are not biased toward the linear model or the versatile model.

In the result section, we added the following sentence “The distributions of probability and confidence reports were also largely similar (see Supp. Fig. 1).”

In the methods section detailing the simulations, we inserted the following sentence: “The distributions of normative probability and confidence reports used in the simulations match, by construction, the ones used in the analysis of the fMRI data.”

Reviewer #3 (Remarks on code availability):

The code was quite well organized. I did not run it but it is one of the better code repos I've seen.

Point-by-Point Response to Reviewers

REVIEWERS' COMMENTS

Reviewer #1 (Remarks to the Author):

The authors have again been very responsive to my comment (and those of the other reviewers) and have changed the introduction and discussion of the results to more precisely characterize the aim of the study and the conclusions that can be drawn from the results. They have also added a code notebook and discuss the utility of their method for studying probability (as opposed to just the implications of their specific findings), which is really useful for the field. I think this version of the manuscript is vastly improved and should be published. It will find many interested readers.

Reviewer #1 (Remarks on code availability):

I have not had time for a very thorough review of the code, but there are clear instructions on how to run it, the code itself is well structured, and it is very useful both for reproducing the figures of the paper and for using this analysis method for other purposes. This is what authors should provide!

Reviewer #2 (Remarks to the Author):

All my comments have been addressed

I would like to thank the authors for their work in responding to my comments

Reviewer #3 (Remarks to the Author):

I thank the authors for their responses, but their response to the last point I raised regarding confidence was rather confusing. First, the authors refer to Supplementary Fig 6 for a histogram of subjective confidence, but that only shows the simulated RSA matrices. I believe the authors are instead referring to Supplementary Fig 1. However, in that figure, the skew in the subjective confidence measures are very noticeable, and only “spans the full range” in the technical sense. The same issue is present for normative confidence. Given this, I don’t see how the authors’ response address the issue of range compression for the confidence measures.

To address this recurring concern and eliminate any potential doubt on the part of both the reviewer and the reader, we have added the following clarification to the final version of our manuscript in the Results section (new text highlighted in red below):

"In other words, an encoding of probability cannot be mistaken for an encoding of confidence, or vice versa. Several other aspects of these simulations are worth noting. Whatever the code, it should be easier to detect correlates of confidence than of probability (because z -R² is larger on average). **Crucially, despite different distributions of confidence and probability during the task (see Supplementary Fig. 1),** the ability to correctly detect a non-monotonic code is high for both. In particular, the simulations show that the task and the analysis do not bias the results in favor of a non-monotonic code for probability rather than for confidence, indicating that the empirical results detailed below do not arise from a methodological artifact."

Reviewer #3 (Remarks on code availability):

No concerns.